# Great Minds Think Alike: The Universal Convergence Trend of Input Salience

**Yipei Wang, Jeffrey Mark Siskind, Xiaoqian Wang**
Elmore Family School of Electrical and Computer Engineering
Purdue University
West Lafayette, IN 47907
`wang4865,qobi,joywang@purdue.edu`

## Abstract

Uncertainty is introduced in optimized DNNs through stochastic algorithms, forming specific distributions. Training models can be seen as random sampling from this distribution of optimized models. In this work, we study the distribution of optimized DNNs as a family of functions by leveraging a pointwise approach. We focus on the input saliency maps, as the input gradient field is decisive to the models' mathematical essence. Our investigation of saliency maps reveals a counter-intuitive trend: two stochastically optimized models tend to resemble each other more as either of their capacities increases. Therefore, we hypothesize several properties of these distributions, suggesting that (1) Within the same model architecture (e.g., CNNs, ResNets), different family variants (e.g., varying capacities) tend to align in terms of their population mean directions of the input salience. And (2) the distributions of optimized models follow a convergence trend to their shared population mean as the capacity increases. Furthermore, we also propose semi-parametric distributions based on the Saw distribution to model the convergence trend, satisfying all the counter-intuitive observations. Our experiments shed light on the significant implications of our hypotheses in various application domains, including black-box attacks, deep ensembles, etc. These findings not only enhance our understanding of DNN behaviors but also offer valuable insights for their practical application in diverse areas of deep learning.

## 1 Introduction

The advancement in computational power has significantly enhanced the capabilities of Deep Neural Networks (DNNs), leading to their unparalleled expressiveness and success in a multitude of applications across various fields (Krizhevsky et al., 2012; He et al., 2016; Rajkomar et al., 2018; Berner et al., 2019; Rombach et al., 2022; Padmaja et al., 2023; Thirunavukarasu et al., 2023). Despite these achievements, DNNs remain enigmatic, not only to end-users but also to researchers and practitioners (Ribeiro et al., 2016; Rudin, 2018; Preece et al., 2018). Due to the over-parameterization nature of modern DNNs, they are capable of reaching zero loss in the training distribution (Goodfellow et al., 2014b; Allen-Zhu et al., 2019; Du et al., 2019). Furthermore, the inherent stochastic nature of training algorithms means that even when using the same training data, DNNs tend to converge to various minima (Huang et al., 2017; Liu et al., 2020). Thus even though these models may exhibit comparable performance in terms of metrics like testing loss or accuracy, their underlying mechanisms can still differ significantly. Because of the stochastic nature of the training procedure, optimized DNNs collectively form a distribution over the functional space $\mathcal{C}^1(\mathcal{X})$, and training DNNs from scratch is thereby equivalent to randomly sampling from such a distribution without any guarantee. This inherent opacity, combined with the high dimensionality and nonlinearity, limits our understanding of the internal mechanisms of DNNs.

38th Conference on Neural Information Processing Systems (NeurIPS 2024).

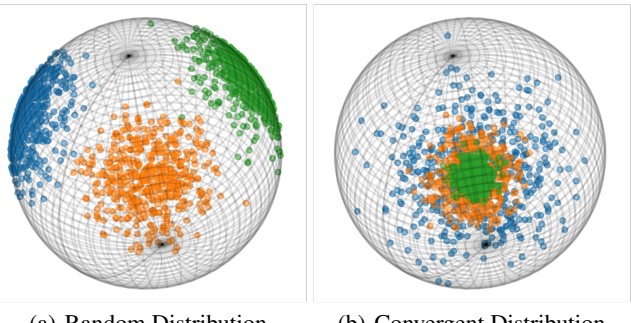

(a) Random Distribution  (b) Convergent Distribution

Figure 1: A synthetic illustration of the distribution of the directional gradients of stochastically optimized models of the same input data. The subfigures demonstrate (a) an intuitive, stochastic scenario, where the distributions of different model families are not closely dependent. and (b) the converging distribution trend introduced by our hypothesis. Different colors represent different model families, and points represent different optimized models.

In response to these challenges, we study the aforementioned distributions. By adopting a pointwise approach, our focus is on the distribution of input salience (Simonyan et al., 2013) from the context of eXplainable Artificial Intelligence (XAI), which aims to demystify the inner workings of these complex models (Gunning and Aha, 2019; Arrieta et al., 2020; Van der Velden et al., 2022). Saliency maps, particularly in the form of *input gradients*, represent the data points within the gradient fields of DNNs. Thus the study of gradients can offer a deterministic view of the landscape of model predictions. This approach allows us to examine the intricate nuances of DNNs in a more structured and analytical manner.

For clarity, in the following context, we distinguish between the term **model architecture** (e.g. skip/direct connections) from the term **model family**. The latter refers to a specific collection of models $\mathcal{F}$, that differ only in capacity as determined by width and depth. Two models are said to be in the same family if they differ only in parameter values. Given an input, varying model families result in distinct distributions. A synthetic visualization of such distributions is shown in Figure 1(a). Different models are depicted by the points. However, the relationship between different model families, represented by various colors, remains elusive. In this work, we introduce and verify several hypotheses to uncover a striking pattern. (1) Within the same model architecture (e.g., CNNs, ResNets), different family variants (e.g., varying capacities) tend to align in terms of their population mean direction. (2) As the model capacities increase, the variance within the distribution of the same family diminishes. This leads to a converging trend of the distributions. Both hypotheses are illustrated in Figure 1(b). Additionally, we introduce a semi-parametric approach to model these distributions, providing detailed quantification of the convergence.

The similarities observed in input salience have direct implications for understanding the important vulnerability of DNNs regarding gradient attacks (Szegedy et al., 2013; Goodfellow et al., 2014a). In particular, in black-box attack settings, the gradients of the target model are not directly accessible. A higher degree of salience similarity naturally enhances transferability (Chen et al., 2023). Our findings elucidate why models with larger capacity consistently exhibit superiority in terms of adversarial robustness compared to smaller models (Madry et al., 2017; Gustafsson et al., 2020; Li et al., 2020; Bubeck and Sellke, 2021). Moreover, given that the mean direction is aligned across different models, it is possible to approximate this mean direction by randomly sampling from a set of independently optimized models. We demonstrate that these estimated mean directions can attain a near-perfect cosine similarity of almost 1.0, even between completely independent models or ensembles, in a high-dimensional space. Moreover, note that deep ensembles essentially calculate this population mean direction (Lee et al., 2015; Lakshminarayanan et al., 2017; Fort et al., 2019; Kondratyuk et al., 2020), where the mean of a group of independently trained models can improve the performance. As a consequence, the insights of our hypotheses also shed light on this phenomenon which, although empirically successful, has been somewhat enigmatic in terms of their source of capability (Lobacheva et al., 2020; Deng and Shi, 2021; Abe et al., 2022; Theisen et al., 2023). Furthermore, since deep ensembles approximate the aligned mean directions much faster than scaling up single models, this

also demystifies the significant black-box attack transferability of deep ensembles (Yang et al., 2021; Chen et al., 2023). Our research thus not only advances the understanding of model behavior in practical applications but also contributes to the broader field of AI trustworthiness and efficiency. Our main contributions can be summarized as follows:

- We reveal an appealing phenomenon where the mean distribution directions of input salience across different model families have extremely high resemblance.
- We empirically demonstrate the distribution converges towards the mean direction as model capacity increases.
- Incorporating both empirical observations and theoretical analysis, we hypothesize distributional properties of optimized models, quantifying the aforementioned phenomena.
- The hypotheses effectively explain many hitherto unclear phenomena such as black-box attack transferability, the efficacy of deep ensemble methods, etc.

## 2 The Convergence of Input Saliency

### 2.1 Salience Similarities

**Notation.** Let $\mathcal{X} \times \mathcal{Y} = \mathcal{D}$ denote the dataset, where $\mathcal{X} \subset \mathbb{R}^d$ is the input set and $\mathcal{Y} = [c]$ is the set of labels and $c \in \mathbb{N}_+$ is the number of classes. Following the benign overfitting phenomenon (Bartlett et al., 2020; Papyan et al., 2020; Cao et al., 2022), we let $\mathcal{F} = \{f | \mathcal{L}(f; \mathcal{X}_{\text{train}}, \mathcal{Y}_{\text{train}}) < 10^{-3}\}$ denote a family of optimized models, distinguished by different architectures, such as vanilla sequential CNNs, skipping blocks, etc. $\mathcal{L}$ denotes the expected cross-entropy loss for the training distribution. For simplicity, we focus on $f : \mathbb{R}^d \to \mathbb{R}$, which predicts the logit specifically for the *targeted class*. This is to stay consistent with XAI methods. We demonstrate in Appendix A that the difference between logit and probability (Wang and Wang, 2022) does not affect the observed phenomena. Unless otherwise indicated, experiments are carried out over the test set $\mathcal{X} = \mathcal{X}_{test}, \mathcal{Y} = \mathcal{Y}_{test}$. Within the same architecture, model capacity is determined by both the width and the depth. Since it is more difficult to model depth as a single variable, we model varying depth as different families $\mathcal{F}$ but model varying width $k$ as a parameter of the family, $\mathcal{F}(k)$.

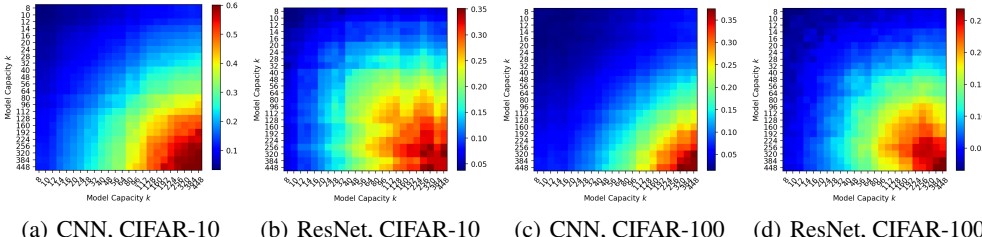

|  (a) CNN, CIFAR-10 | (b) ResNet, CIFAR-10 | (c) CNN, CIFAR-100 | (d) ResNet, CIFAR-100 |

Figure 2: The individual similarity $\rho_{ind}(f^{(1)}, f^{(2)}) = \mathbb{E}_{\boldsymbol{x} \in \mathcal{X}}[\texttt{CosSim}(\nabla_{\boldsymbol{x}} f^{(1)}(\boldsymbol{x}), \nabla_{\boldsymbol{x}} f^{(2)}(\boldsymbol{x}))]$, where $f^{(1)} \in \mathcal{F}(k_1), f^{(2)} \in \mathcal{F}(k_2)$. CIFAR-10/100 and CNN & ResNets are tested here.

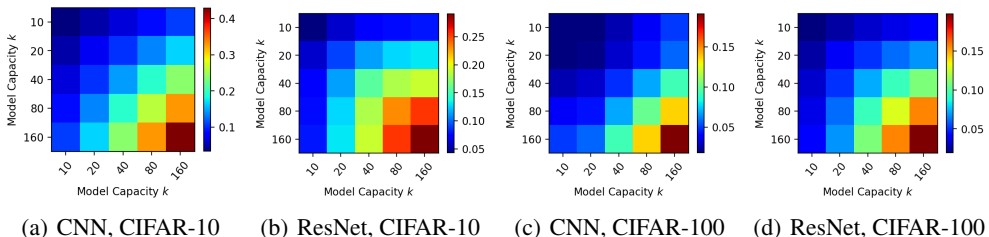

|  (a) CNN, CIFAR-10 | (b) ResNet, CIFAR-10 | (c) CNN, CIFAR-100 | (d) ResNet, CIFAR-100 |

Figure 3: The expected similarity $\rho(k_1, k_2)$ between model families of varying capacities $k_1, k_2$. Here the datasets are CIFAR-10/100, and the models are CNN and ResNets.

**The Increasing Similarity.** Let $\mathtt{CosSim} : \mathbb{R}^d \times \mathbb{R}^d \to [-1, 1]$ denote the cosine similarity metric, then the individual similarity between the input salience of two given models $f^{(1)} \in \mathcal{F}(k_1), f^{(2)} \in \mathcal{F}(k_2)$ given input $\boldsymbol{x}$ is $\rho_{ind}(f^{(1)}, f^{(2)}; \boldsymbol{x}) = \mathtt{CosSim}(\nabla_{\boldsymbol{x}} f^{(1)}(\boldsymbol{x}), \nabla_{\boldsymbol{x}} f^{(2)}(\boldsymbol{x}))$. Taking the entire testing set into consideration, denote $\rho_{ind}(f^{(1)}, f^{(2)}) = \mathbb{E}_{\boldsymbol{x} \in \mathcal{X}}[\rho_{ind}(f^{(1)}, f^{(2)}; \boldsymbol{x})]$. In Figure 2, the expectations over the testing set $\mathbb{E}_{\boldsymbol{x} \in \mathcal{X}} \mathtt{CosSim}(\nabla_{\boldsymbol{x}} f^{(1)}(\boldsymbol{x}), \nabla_{\boldsymbol{x}} f^{(2)}(\boldsymbol{x}))$ with varying $k_1, k_2 \in K$ are illustrated. Here, we define $K = \{j2^i : 4 \le j \le 7, 1 \le i \le 6\} = \{8, 10, 12, 14, 16, 20, \cdots, 384, 448\}$ to balance between finer linear scaling and coarser exponential scaling. It can be observed that the similarity between two stochastically optimized models $f^{(1)}, f^{(2)}$ has an increasing trend with respect to both $k_1, k_2$. Two different architectures CNN and ResNet are included. To rule out the influence of any single model, we define the similarity between families $\mathcal{F}(k_1), \mathcal{F}(k_2)$ for a given input $\boldsymbol{x}$ by taking the expectation of the two models:

$$\rho(k_1, k_2; \boldsymbol{x}) := \mathbb{E}_{f^{(1)} \in \mathcal{F}(k_1), f^{(2)} \in \mathcal{F}(k_2)} \mathtt{CosSim}(\nabla_{\boldsymbol{x}} f^{(1)}(\boldsymbol{x}), \nabla_{\boldsymbol{x}} f^{(2)}(\boldsymbol{x})) \tag{1}$$

The global similarity between models of widths $k_1, k_2$ is then denoted by $\rho(k_1, k_2) = \mathbb{E}_{\boldsymbol{x} \in \mathcal{X}} \rho(k_1, k_2; \boldsymbol{x})$. Note that estimating this value requires training numerous $f \in \mathcal{F}(k)$ for each $k \in K$. Therefore, we carry out the experiments over $K' = \{j2^i : j = 5i = 1, 2, 3, 4, 5\} = \{10, 20, 40, 80, 160\} \subset K$. For each $k_1, k_2 \in K'$, 100 models are sampled respectively to empirically estimate the expectation over $\mathcal{F}(k_1), \mathcal{F}(k_2)$. As observed in Figure 3, $\rho(k_1, k_2)$ has an increasing trend w.r.t. both $k_1, k_2$. Compared with Figure 3, the average similarity over the model families is similar to the individual cosine similarity for the same $k$ values. As a result, studying the similarity of two randomly sampled models instead of the expectation over $\mathcal{F}$s can significantly alleviate the computational burden. This is further discussed in detail in Section 3.2. Besides, It trivially follows that $\forall k_1 > k_2, \rho(k_1, k_1) > \rho(k_1, k_2) > \rho(k_2, k_2)$, which suggests that *larger models tend to resemble smaller models even more than smaller models themselves.* – Even if the two smaller models only differ in the random seeds during training. Please refer to Appendix A for the results of more datasets, where such increasing trends still exist.

## 2.2 The Spherical Distribution of the Salience

Since the cosine similarity can be written as the inner product between $\frac{\nabla_{\boldsymbol{x}} f^{(1)}(\boldsymbol{x})}{\|\nabla_{\boldsymbol{x}} f^{(1)}(\boldsymbol{x})\|}$ and $\frac{\nabla_{\boldsymbol{x}} f^{(2)}(\boldsymbol{x})}{\|\nabla_{\boldsymbol{x}} f^{(2)}(\boldsymbol{x})\|}$, which are high-dimensional unit vectors, we explore the properties and potential distributions of $\mathcal{F}$ through the perspective of spherical statistics. Given an input $\boldsymbol{x} \in \mathcal{X}$, we denote by $\mathcal{G}_k(\boldsymbol{x})$ the set of all possible gradient directions of input $\boldsymbol{x}$ regarding the models $f$ in $\mathcal{F}(k)$. Formally, let

$$\mathcal{G}_k(\boldsymbol{x}) = \{\boldsymbol{u} = \nabla_{\boldsymbol{x}} f(\boldsymbol{x}) / \|\nabla_{\boldsymbol{x}} f(\boldsymbol{x})\| \big| \forall f \in \mathcal{F}(k)\}, \forall \boldsymbol{x} \in \mathcal{X} \tag{2}$$

Then the similarity is re-written as the inner product $\rho(k_1, k_2; \boldsymbol{x}) = \mathbb{E}_{\boldsymbol{u}_1 \in \mathcal{G}_{k_1}(\boldsymbol{x}), \boldsymbol{u}_2 \in \mathcal{G}_{k_2}(\boldsymbol{x})}[\boldsymbol{u}_1^T \boldsymbol{u}_2]$.

**The Intra-Family vs. Cross-Family Paradox.** An interesting paradox is raised as $\rho(\cdot, \cdot)$ increases with both inputs. Naturally, one would reasonably deduce that two models $f^{(1)}, f^{(2)} \in \mathcal{F}(k_1)$ should resemble each other since they are from the same family (i.e. having the exact same structure and only differ in training seeds). However, since $\rho(k_2, k_1) > \rho(k_1, k_1)$, the cross-model family similarity becomes greater than the intra-model family similarity. To uncover the mystery of the observations, we present the intuitive understanding and the rigorously analyzed hypotheses as follows.

**The Intra-Family Hypothesis.** Note that for intra-family scenario, $\boldsymbol{u}, \boldsymbol{v} \in \mathcal{G}_k(\boldsymbol{x})$ are i.i.d., the similarity can be written as $\rho(k, k; \boldsymbol{x}) = (\mathbb{E}_{\mathcal{G}_k(\boldsymbol{x})}[\boldsymbol{u}])^T (\mathbb{E}_{\mathcal{G}_k(\boldsymbol{x})}[\boldsymbol{v}]) = \|\mathbb{E}_{\mathcal{G}_k(\boldsymbol{x})}[\boldsymbol{u}]\|_2^2$, which denotes the square of the *population mean resultant length* (Mardia et al., 2000) of $\mathcal{G}_k(\boldsymbol{x})$. The population mean resultant length $\sqrt{\rho(k, k; \boldsymbol{x})}$ quantifies the degree of dispersion of $\mathcal{G}_k(\boldsymbol{x})$, where a larger length suggests a more concentrated distribution. In directional statistics, the degree of dispersion is usually quantified by the spherical variance $2(1 - \sqrt{\rho(k, k; \boldsymbol{x})})$ or the total variation $1 - \rho(k, k; \boldsymbol{x})$. Since $\rho(k, k; \boldsymbol{x})$ also increases w.r.t. $k$, this suggests an increasing concentration of input salience of models as the width $k$ of the model increases. In conclusion, *the larger the models are, the smaller the spherical variance of the salience is.* Formally, we propose the following hypothesis.

- **Hypothesis I (H1)**: Let $k$ denote the width (capacity) of the model and $\mathcal{G}_k(\boldsymbol{x}) = \{\boldsymbol{u} = \frac{\nabla_{\boldsymbol{x}} f(\boldsymbol{x})}{\|\nabla_{\boldsymbol{x}} f(\boldsymbol{x})\|} | \forall f \in \mathcal{F}(k)\}$ denote the set of input gradient directions regarding $\boldsymbol{x}$. Then

$\mathbb{E}_{\mathcal{G}_k(\boldsymbol{x})}[\boldsymbol{u}] = \sqrt{\rho(k,k;\boldsymbol{x})}\boldsymbol{\mu}(k;\boldsymbol{x})$ and $\rho(k,k;\boldsymbol{x})$ *increases with* $k$. Here $\boldsymbol{\mu}(k,\boldsymbol{x}) = \frac{\mathbb{E}_{\mathcal{G}_k(\boldsymbol{x})}[\boldsymbol{u}]}{\|\mathbb{E}_{\mathcal{G}_k(\boldsymbol{x})}[\boldsymbol{u}]\|}$ denotes the unit mean direction of $\mathcal{G}_k(\boldsymbol{x})$.

Note that *H1 also holds for the change of model depths*, which is positively related to the dispersion degree of the distribution. However, the change in model depth inevitably affects model width. Thus, we only provide empirical verification in Section 3 but do not include it as a part of H2.

**The Cross-Family Hypothesis.** Unlike the intra-family similarity, the increasing cross-family similarity is where the phenomenon becomes counter-intuitive. Then due to the increasing intra-family similarities, when $k_1, k_2$ increase, $\boldsymbol{u}_1 \in \mathcal{G}_{k_1}(\boldsymbol{x})$ becomes closer to $\boldsymbol{\mu}(k_1;\boldsymbol{x})$, $\boldsymbol{u}_2 \in \mathcal{G}_{k_2}(\boldsymbol{x})$ becomes closer to $\boldsymbol{\mu}(k_2;\boldsymbol{x})$. However, the cross-familty similarities suggest that as $\boldsymbol{u}_1, \boldsymbol{u}_2$ approach their respective mean directions, their similarity increases as well. This indicates that the mean directions $\boldsymbol{\mu}(k_1;\boldsymbol{x}), \boldsymbol{\mu}(k_2;\boldsymbol{x})$ are similar, too. Formally, this intuition is considered as follows. For $k_1 > k_2$, when $\boldsymbol{\mu}(k_1;\boldsymbol{x})$ and $\boldsymbol{\mu}(k_2;\boldsymbol{x})$ are sufficiently similar, $\boldsymbol{\mu}(k_1;\boldsymbol{x})^T\boldsymbol{\mu}(k_2;\boldsymbol{x}) \approx \|\boldsymbol{\mu}(k_1;\boldsymbol{x})\|\|\boldsymbol{\mu}(k_1;\boldsymbol{x})\|$. Thus we have

$$\begin{aligned}
\rho(k_1, k_2; \boldsymbol{x}) =&\mathbb{E}_{\boldsymbol{u}_1 \in \mathcal{G}_{k_1}(\boldsymbol{x}), \boldsymbol{u}_2 \in \mathcal{G}_{k_2}(\boldsymbol{x})}[\boldsymbol{u}_1^T\boldsymbol{u}_2] = \mathbb{E}_{\boldsymbol{u}_1 \in \mathcal{G}_{k_1}(\boldsymbol{x})}[\boldsymbol{u}_1]^T \mathbb{E}_{\boldsymbol{u}_2 \in \mathcal{G}_{k_1}(\boldsymbol{x})}[\boldsymbol{u}_2] \\
\approx &\|\mathbb{E}_{\boldsymbol{u}_1 \in \mathcal{G}_{k_1}(\boldsymbol{x})}[\boldsymbol{u}_1]\|\|\mathbb{E}_{\boldsymbol{u}_2 \in \mathcal{G}_{k_1}(\boldsymbol{x})}[\boldsymbol{u}_2]\| = \sqrt{\rho(k_1, k_1; \boldsymbol{x})\rho(k_2, k_2; \boldsymbol{x})},
\end{aligned} \quad (3)$$

which is monotonic w.r.t. both $k_1, k_2$. Formally, this is summarized as

- **Hypothesis II (H2)**: Let $\mathcal{G}_{k_1}(\boldsymbol{x}), \mathcal{G}_{k_2}(\boldsymbol{x})$ denote the input gradient directions of two model families where $k_1 \neq k_2$. Then $\boldsymbol{\mu}(k_1;\boldsymbol{x}) \approx \boldsymbol{\mu}(k_2;\boldsymbol{x})$ regardless of $k_1, k_2$.

The two hypotheses are both empirically verified. For a smoother flow of the presentation, we defer the detailed experiments to Section 3. The basic ideas of H1 and H2 are illustrated in Figure 4 (a).

## 2.3 The Directional Distribution of Gradients

Given the analysis and hypothesis above, one can have an overview of the models' internal mechanisms. As the model capacity increases, models are distributed in a more concentrated manner, while the mean direction stays almost invariant. To better understand the models' behavior with the stochasticity, we delve into the distribution of $\mathcal{G}_k(\boldsymbol{x})$ and present a semi-parametric analysis with experimental verification. A general form of centralized symmetric distribution over hypersphere is known as the Saw distribution (Fisher et al., 1993) $p(\boldsymbol{u}; \boldsymbol{\mu}) = \frac{\psi(\boldsymbol{u}^T\boldsymbol{\mu})}{Z}$, where $\boldsymbol{\mu}$ is the mean direction with $\|\boldsymbol{\mu}\| = 1$, $\psi \in \mathcal{C}([-1, 1])$, and $Z = \int_{S^{d-1}} \psi(\boldsymbol{u}^T\boldsymbol{\mu})\mathrm{d}\boldsymbol{u}$ is the normalization term for distributions. Due to the symmetry assumption, the shape of the distribution is solely determined by $\psi$. For example, a monotonically increasing $\psi$ suggests that $\boldsymbol{u}$ is distributed more densely near the mean direction and sparsely distant from the mean direction. Considering the concentration trend of gradients, we hypothesize that $\psi_k(\cdot)$ of $\mathcal{G}_k(\boldsymbol{x})$ not only monotonically increases with the input, but also increases faster with greater $k$ values.

**Marginalization.** For $\forall \boldsymbol{u} \in \mathcal{G}_k(\boldsymbol{x})$, it can be decomposed to $\boldsymbol{u} = t \cdot \boldsymbol{\mu}(\boldsymbol{x}) + \sqrt{1 - t^2} \cdot \boldsymbol{\mu}(\boldsymbol{x})^\perp$, where $\boldsymbol{\mu}(\boldsymbol{x})^\perp$ is a unit tangent to $S^{d-1}$ at $\boldsymbol{\mu}$. Then $t = \boldsymbol{u}^T\boldsymbol{\mu}(\boldsymbol{x})$. This is shown in Figure 4 (b). Note that $\boldsymbol{\mu}(\boldsymbol{x})^\perp$ is independent from $t$, then the distribution of $t$ is the marginal distribution over the intersection between $S^{d-1}$ and the hyperplane spanned by $\boldsymbol{\mu}(\boldsymbol{x})^\perp$, which is a $(d-2)$-dimensional hypersphere. According to the symmetry assumption of Saw distribution, conditioned on a fixed similarity $t$, the distribution of $\boldsymbol{u}|t$ over the dashed $S^{d-2}$ does not affect $\psi$. Therefore, we focus on the marginalized distribution of $t$. Note that the radius of the intersection $S^{d-2}$ is $\sqrt{1 - t^2}$, we thus have $\mathrm{d}\boldsymbol{u} = \frac{2\pi^{(d-1)/2}(1-t^2)^{(d-3)/2}}{\Gamma((d-1)/2)}\mathrm{d}t$, where the density of $t$ is observed by the integral over the corresponding $(d-2)$-hypersphere. As a result, the marginal distribution of $t$ has the PDF

$$p_k(t; \boldsymbol{x}) = \psi_k(t; \boldsymbol{x})(1 - t^2)^{(d-3)/2}/Z' \quad (4)$$

where $Z'$ is a constant normalization term. Note that $(1 - t^2)^{(d-3)/2}$ is a symmetric bell curve centered at $t = 0$. Equation (4) can thus be viewed as using $\psi_k(t; \boldsymbol{x})$ to reweight its PDF $p_{\mathrm{origin}}(t) = (1 - t^2)^{(d-3)/2}\frac{\Gamma(d/2)}{\sqrt{\pi}\Gamma((d-1)/2)}$. Note that here $p_k(t; \boldsymbol{x})$ is the distribution of $t = \boldsymbol{u}^T\boldsymbol{\mu}(k; \boldsymbol{x})$, which can be empirically estimated, the shape of the function $\psi_k$ becomes empirically accessible with varying $k$ values. The empirical studies and verification are provided in Section 3.3.

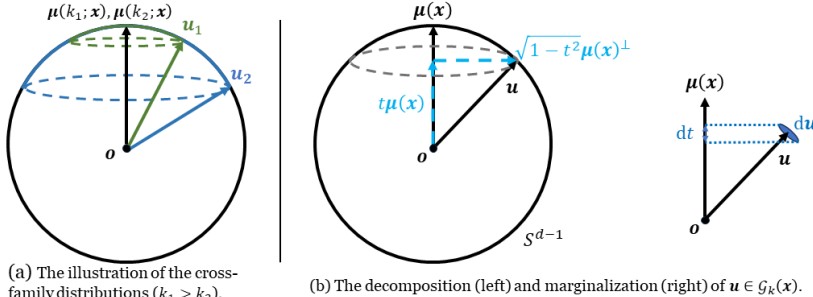

(a) The illustration of the cross-family distributions ($k_1 > k_2$).

(b) The decomposition (left) and marginalization (right) of $\boldsymbol{u} \in \mathcal{G}_k(\boldsymbol{x})$.

Figure 4: (a) presents an illustration of H1 and H2. Blue and green caps represent $\boldsymbol{u}_1 \in \mathcal{G}_{k_1}(\boldsymbol{x})$ (green) and $\boldsymbol{u}_2 \in \mathcal{G}_{k_2}(\boldsymbol{x})$ (blue) regions [2]. H1: larger $k$s lead to smaller spherical variances; H2: the mean directions are extremely similar. (b) illustrates (left) the decomposition of $\boldsymbol{u}$ to the mean direction and the orthogonal direction; and (right) the marginalization of the distribution from $\boldsymbol{u}$ to $t$.

## 3 Empirical Verification of Hypotheses

In this section, we provide comprehensive experimental results to verify the aforementioned hypotheses. First, we introduce the detailed setups of our experiments. They are carried out on Intel(R) Core(TM) i9-9960X CPU @ 3.10GHz with Quadro RTX 6000 GPUs.

**Datasets & Models.** Due to the massive size of experiments, here we mainly follow the setups of the benign overfitting (Nakkiran et al., 2021), which also present a comprehensive study of optimized DNNs through CIFAR-10 and CIFAR-100 (Krizhevsky et al., 2009). Besides, we also include TinyImagenet-200 (Le and Yang, 2015) as a compromise between the computational efficiency and the dataset complexity. As for models, we include CNNs and ResNets as in (Nakkiran et al., 2021). These two models represent the two typical architectures – the direction connection and the skip connection. We also notice a striking capacity gap between them in the original implementation. Therefore, we term them CNNSmall (CS) and ResNetLarge (RL), respectively, and include CNNLarge (CL), and ResNetSmall (RS) to fill the gap. The comparison between the small and large families also shows the influence of depths. As for the training process, following Nakkiran et al. (2021), we use stochastic gradient descent (SGD) as the solver, with a batch size of 128. The input data are normalized, but not augmented. We start with the initial learning rate $\gamma_0 = 0.1$ and update it with $\gamma_t = \gamma_0/\sqrt{1+t}$, where $t$ is the epoch. Please refer to Appendix B for more experimental details.

### 3.1 The Mean Direction Similarity (H2)

H2 can be verified independently from H1 and can provide simplifications and insights to verifying H1. We hence focus on H2 first. As stated in H2, the mean directions of different model families are consistently aligned, i.e., $\boldsymbol{\mu}(k_1; \boldsymbol{x}) \approx \boldsymbol{\mu}(k_2; \boldsymbol{x})$. For each family $\mathcal{F}(k)$ where $k \in K'$, $M = 100$ models with different random seeds are trained. The population mean is then estimated through

$$\tilde{\boldsymbol{\mu}}(k; \boldsymbol{x}) = \Big(\frac{1}{M}\sum_{i=1}^{M}\frac{\nabla_{\boldsymbol{x}}f_i(\boldsymbol{x})}{\|\nabla_{\boldsymbol{x}}f_i(\boldsymbol{x})\|}\Big)\Big/\Big\|\frac{1}{M}\sum_{i=1}^{M}\frac{\nabla_{\boldsymbol{x}}f_i(\boldsymbol{x})}{\|\nabla_{\boldsymbol{x}}f_i(\boldsymbol{x})\|}\Big\| \approx \boldsymbol{\mu}(k; \boldsymbol{x}), f_i \in \mathcal{F}(k). \quad (5)$$

Then the similarity of mean directions are naturally $(\tilde{\boldsymbol{\mu}}(k_1; \boldsymbol{x}))^T \tilde{\boldsymbol{\mu}}(k_2; \boldsymbol{x})$. Note that when $k_1 = k_2$, the 100 models are partitioned by the seeds to avoid overlapping. The results of the expectation over $\mathcal{X}$ are visualized in Figure 5. It can be found that the mean directions have extremely high resemblance within each architecture, as proved by the high cosine similarities. It should be noted that with high dimensionality (e.g. $d = 3072$ for CIFAR), a cosine similarity close to 1 is an extremely significant result. We demonstrate this with the uniform distribution on the hypersphere in Appendix D. The observations verify the hypothesis all $\mathcal{G}_k(\boldsymbol{x})$ almost share the same mean direction within the model architecture. This not only hold across different widths determined by $k$, but also holds across different depths (i.e. CS vs. CL, RS vs. RL). Therefore, the mean direction is mostly related to the certain model architecture instead of any single model $f \in \mathcal{F}$, making it an intrinsic

---

[2]The caps are to illustrate the variance differences. Actual distributions are over the entire hypersphere.

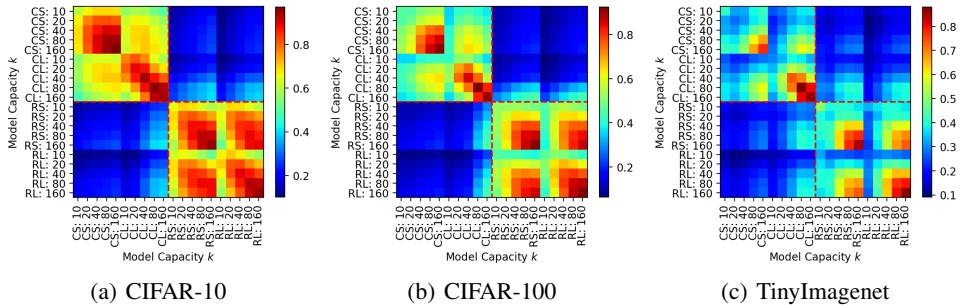

| (a) CIFAR-10 | (b) CIFAR-100 | (c) TinyImagenet |

Figure 5: The heatmap visualization between the estimated population mean directions from different model families. Each entry is computed by $\mathbb{E}_{\boldsymbol{x}\in\mathcal{X}}\texttt{CosSim}(\tilde{\boldsymbol{\mu}}_j(\boldsymbol{x};\mathcal{F}),\tilde{\boldsymbol{\mu}}_{j'}(\boldsymbol{x};\mathcal{F}'))$ The results are generated from CIFAR-10/100 and TinyImagenet datasets. $\mathcal{F} \in$ {CS, CL, RS, RL}.

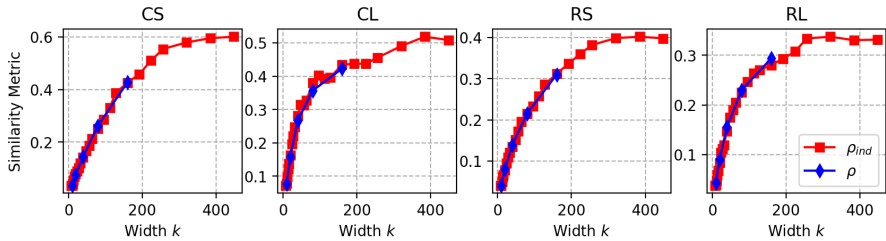

Figure 6: Illustration of (red) $\rho_{ind}(f^{(1)}, f^{(2)})$, $f^{(1)}, f^{(2)} \in \mathcal{F}(k)$ and (blue) $\rho(k, k)$ on CIFAR-10.

property of the model architecture. With this property, how different model architectures differ in mechanisms can be studied by looking deeper into the population mean direction of saliency maps. For instance, it can be observed that the ResNet architecture admits a closer relation between models of different depths, compared with the CNN architecture.

## 3.2   The Decreasing Spherical Variance with $k$ (H1)

**Expectation over $\mathcal{F}$ vs. Conditioned on $f \in \mathcal{F}$.**   As previously discussed, the spherical variance of distributions over the hypersphere can be measured by the population mean resultant length $\sqrt{\rho(k, k; \boldsymbol{x})}$, which, unfortunately, requires an estimation of the mean directions. This can be expensive to study for a comprehensive set $K$ of $k$ values. The experiments on a subset $K' = \{10, 20, 40, 80, 160\}$ are already carried out in Figure 3, shown as the diagonal elements. As $k$ increases, the resultant length increases monotonically, indicating a decreasing spherical variance and a more concentrated distribution around the mean directions.

The computational burden of taking the expectation over $\mathcal{F}$ can be alleviated by considering randomly picked $f$. In order to compare $\rho$ and $\rho_{ind}$, we consider the model-dependent set $\mathcal{S}(f) = \{\rho_{ind}(f, f; \boldsymbol{x}) : \boldsymbol{x} \in \mathcal{X}\}$ for each $f \in \mathcal{F}$. Here we compute the expected Wasserstein distance $\mathbb{E}_{f^{(1)}, f^{(2)} \in \mathcal{F}(k)}\texttt{WD}(\mathcal{S}(f^{(1)}), \mathcal{S}(f^{(2)}))$. This is estimated by the $\binom{M}{2}$ distinct pairs of models. The distances of all 60 (dataset, model family) pairs lie below 0.035. Such observation suggests that after taking the expectation over $\mathcal{X}$, the differences across individual models can be mitigated. Please refer to Table 1 for comprehensive results on all model families and datasets. As a consequence, it suffices to use $\rho_{ind}(f, f)$ for some $f \in \mathcal{F}(k)$ to approximate $\rho(k, k)$. This is in fact the diagonal elements of Figure 2. A comparison between the diagonal elements $\rho_{k,k}$ and $\rho_{ind}(f^{(1)}, f^{(2)})$, $f^{(1)}, f^{(2)} \in \mathcal{F}(k)$ over CIFAR-10 is presented in Figure 6. Please refer to the appendix for other datasets. $\rho_{ind}$ is evaluated over $k \in K$, while $\rho$ is evaluated over $k \in K' \subset K$. It can be found that after taking the average over $\mathcal{X}$, even though $\rho$ is a little smoother than $\rho_{ind}$, they are very consistent. This verifies that the resultant length increases with $k \in K$ in a much more comprehensive set of models. Thus H1 is empirically verified.

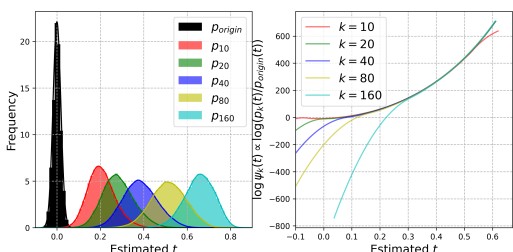
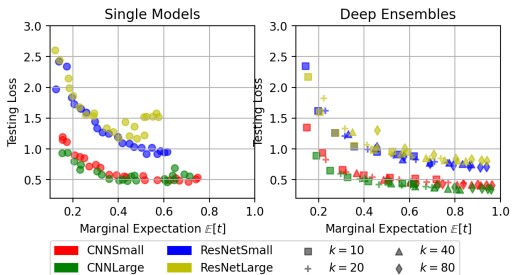

Figure 8: (left) The illustration of the frequency of the mixture $\mathcal{T}_k$, where $k \in \{10, 20, 40, 80, 160\}$. Specifically, the black histogram represents the distribution $p_{origin}$. The dashed curves are the approximated PDF $p_k$ obtained by KDE. The results are generated using CNNSmall and CIFAR-10. (right) The illustration of $\log \frac{p_k}{p_{\text{origin}}}$, which is linearly related to $\log \psi_k$.

Figure 9: The illustration of the relation between the expected testing loss $\mathbb{E}_{\mathcal{X}}[\mathcal{L}]$ and the marginal expectation $\mathbb{E}_{\mathcal{X}}[t]$, where models are from (a) single models with varying capacities (b) deep ensembles with varying member #. Each color represents a model family. In particular, in (b), different marker shapes indicate different $k \in [10, 20, 40, 80]$ of the ensembles.

## 3.3 The Shape of the Saw Distribution

Given $\mathcal{G}_k(\boldsymbol{x})$, the marginalized distribution $p_k(t; \boldsymbol{x})$ can be approximated by $\mathcal{T}_k(\boldsymbol{x}) = \{\tilde{t} = \boldsymbol{u}^T \tilde{\boldsymbol{\mu}}(k; \boldsymbol{x}) | \boldsymbol{u} \in \mathcal{G}_k(\boldsymbol{x})\}$. In order to obtain the global results over test dataset $\mathcal{X}$, consider the unions of different input samples $\mathcal{T}_k = \cup_{\boldsymbol{x} \in \mathcal{X}} \mathcal{T}_k(\boldsymbol{x})$. This is an approximation to the mixture distributions $p_k(t) = \frac{1}{|\mathcal{X}|} \sum_{\boldsymbol{x} \in \mathcal{X}} p_k(t; \boldsymbol{x})$. We plot the histogram of $\mathcal{T}_k$ with $k \in K'$ for CNNSmall and CIFAR-10 in Figure 8. The left figure shows $p_{origin}$ and the estimated $p_k$, visualized by different colors. Qualitatively, $p_k(t)$ has higher means with greater $k$ values. The reason why the density of $p_k$ is not centered at $t = 1$ (i.e. $\boldsymbol{u} = \boldsymbol{\mu}$) is because as $t$ increases, the size of the $(d-2)$-hypershpere decreases with $p_{\text{origin}}(t) = (1-t^2)^{(d-3)/2} \frac{\Gamma(d/2)}{\sqrt{\pi}\Gamma((d-1)/2)}$, which is shown as black histograms. This is much faster than the increase of $\psi_k$. The shape of $\psi_k$ is determined by $p_k/p_{\text{origin}}$ with normalization. From the right figure, by comparing $p_k$ with $p_{\text{PDF}}$, it can be empirically verified that $\psi_k(t)$ is increasing vastly. It is also observed that larger $k$s lead to a faster increase of $\psi_k$ and higher $\mathbb{E}[t]$. This also provides a quantitative understanding of H1 and H2. The results of other model architectures and datasets can be found in the appendix.

**Verification of the Symmetry.** Saw distributions study the marginalized value $t = \boldsymbol{u}^T \boldsymbol{\mu}$ to directly focus on the degree of concentration of the gradients. This naturally leads to rotationally symmetric distributions since the distribution on the intersection between $S^{d-1}$ and the hyperplane does not affect the distribution of $t$. We thus carry out an empirical study of the distribution on the intersection (i.e. conditioned on $t$). Specifically, we train 1000 CNN models with $K = 40$ and seeds 1-1000 on CIFAR-10 and compute $t$ regarding each test sample. The distribution of the first sample is visualized in Figure 7. We partition the range of $t$ into 10 intervals by every 10 percent of the frequency, and inspect the direction of the mean of the gradients in each interval, each direction is estimated by 100 models. If these conditional mean directions are consistent with the population mean direction, then the gradients are symmetrically distributed on each $S^{d-2}$ hypersphere (R7(right)), thus

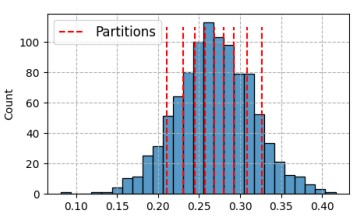

Figure 7: The marginal distribution of $t$ of the first test sample of CIFAR-10. Red dashed lines partition the range of $t$ every 10 percent of the frequency.

verifying the rotational symmetry. We investigate the cosine similarities between the conditional and unconditional mean directions on the first 1000 samples. The $10 \times 1000$ similarity values have a mean and std at approximately 0.970 and 0.013 respectively. Thus the rotational symmetry is empirically verified.

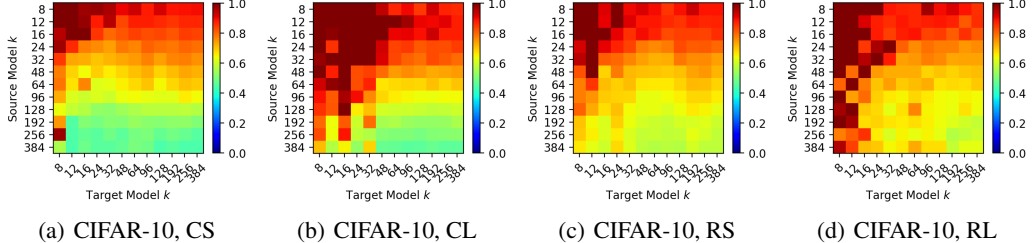

| (a) CIFAR-10, CS | (b) CIFAR-10, CL | (c) CIFAR-10, RS | (d) CIFAR-10, RL |

Figure 10: The results of single model black-box attack. The value of each entry is $\alpha(k_1, k_2)$ for different model capacities, where $k_1$ is the width parameter of the source model and $k_2$ is the width parameter of the target model.

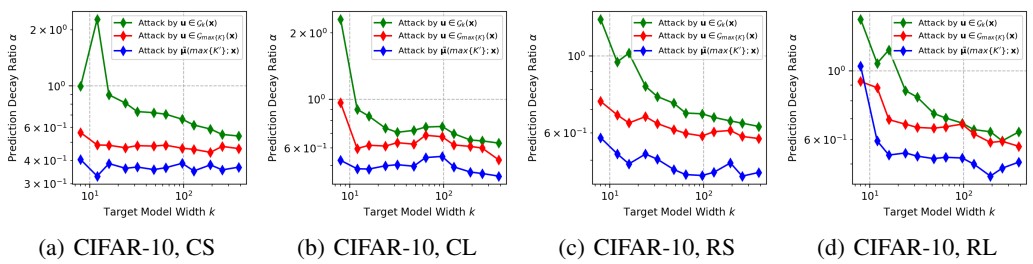

| (a) CIFAR-10, CS | (b) CIFAR-10, CL | (c) CIFAR-10, RS | (d) CIFAR-10, RL |

Figure 11: The comparison between the single-model attack from the largest model (red), the single-model attack from the very same capacity (green) and the attack by the mean direction (blue).

## 4 Applications of Hypotheses

### 4.1 Deep Ensemble: Why Does It Work?

After verifying the hypotheses, we explore possible applications and implications of the discovered phenomena. The deep ensemble method makes use of the stochasticity to of models by incorporating the predictions from $m$ members. While deep ensembles have been verified to be effective in improving performance, the source of such capability remains mysterious. Note that ensemble members are i.i.d. optimized models with SGD, which correspond to the population mean of our hypothesis. We thus provide another perspective in understanding the capability of deep ensembles.

For single models, as the model capacity increases, benign overfitting suggests that the testing loss decreases, too. We deduce that this is because the distribution of larger models becomes more concentrated, and combined with H2, the closeness to the aligned population mean is highly related to the models' testing performance. As shown in Figure 9(a), it can be observed that the expected loss $\mathbb{E}_\mathcal{X}[\mathcal{L}]$ and the marginal expectation $\mathbb{E}_\mathcal{X}[t]$ are highly correlated. Similarly, deep ensemble approximates the population mean much more effectively by increasing the number $m$ of members than scaling up a single model by $k$. We thus scale up the deep ensemble by changing the number of ensemble members. The results are shown in Figure 9(b). It can be found that (1) the correlation between $\mathbb{E}_\mathcal{X}[\mathcal{L}]$ and $\mathbb{E}_\mathcal{X}[t]$ is not only significant, and (2) the correlation pattern is shared between two completely different scaling mechanisms, single model scaling and model ensembles.

### 4.2 Black-Box Attack via Saliency Similarity

The understanding of adversarial attacks can benefit from the behaviors of the input salience of optimized models given their close relation to input gradients. We verify the aforementioned similarities through the black-box attacks, where the adversarial samples are generated from the gradients of source models while the gradients of the target models are not available. Let $f^{(1)} \in \mathcal{F}(k_1)$ denote the source model and $f^{(2)} \in \mathcal{F}(k_2)$ denote the target model. We define the attack rate from

$f^{(1)}$ to $f^{(2)}$ similar to $\rho_{ind}$ as

$$\alpha(f^{(1)}, f^{(2)}) = \mathop{\mathbb{E}}_{\boldsymbol{x} \in \mathcal{X}} \left[ f^{(2)}(\boldsymbol{x} - \epsilon \cdot \texttt{sign}(\nabla_{\boldsymbol{x}} f^{(1)}(\boldsymbol{x}))) / f^{(2)}(\boldsymbol{x}) \right] \tag{6}$$

which is the performance decay of $f^{(2)}$ when attacked by model $f^{(1)}$. Small $\alpha(k_1, k_2)$ values suggest successful attack from $f^{(1)}$ to $f^{(2)}$. The results are shown in Figure 10, where the attack step is set to $\epsilon = 0.05$. In each heatmap figure, the $y$-axis represents the width of the source models, while the $x$-axis represents the width of the target models. It can be observed that larger models succeed in attacking smaller models, but the opposite is not true. To attack a large model, the gradient needs to be generated from a model of a comparable level in terms of capacity.

**Mean Direction Attack.** According to the verified hypothesis H2, for any two individual models $f^{(1)} \in \mathcal{F}(k_1), f^{(2)} \in \mathcal{F}(k_2)$ the mean directions $\boldsymbol{\mu}(k; \boldsymbol{x})$ is closer to both of them than themselves, regardless of $k, k_1, k_2$. It is then suggested that using the mean gradient can perform more successful black-box attacks. We employ the mean direction $\tilde{\boldsymbol{\mu}}(160, \boldsymbol{x})$ to attack models of different capacities, and compare the results with the attack from the largest single model (red) and the attack from the models of the identical structure (green). The results are shown in Figure 11, where it can be observed that the mean direction of salience transfers much more successfully than single models.

# 5 Conclusions

In this paper, we introduce hypotheses to explain the observations on the input salience convergence w.r.t. the model capacities. Under the same model architecture, stochastic algorithms such as SGD, result in certain distributions of optimized models. We hypothesize and use pointwise methods to verify that such distribution follows a Saw distribution with **aligned population means**, which is invariant from the model families. Besides, the variance of the distribution decreases as the model capacity increases, suggesting a convergence trend of the models' internal mechanism – the larger the models are, the less variant they tend to be affected by the randomness from the stochastic algorithm during the training phase. Furthermore, since the distributions converge towards the aligned population mean direction, the limiting points can be estimated by the population mean of models. Based on this, we present comprehensive experiments on the properties of the limiting model and demonstrate its capability in various domains, such as the black-box attack transferability, and the explanation of the effectiveness of deep ensembles. However, it is admitted that, due to the high computational burden, although improved from CIFAR-10/100 to TinyImagenet compared to (Nakkiran et al., 2021), our experiments are limited to rather small datasets.

Our introduced hypotheses also lead to various interesting topics. Note that the aligned mean direction stays invariant to the model families, which indicates such population mean is more related to the essence of the dataset itself rather than any single model. Leveraging this property can bring a deeper and more comprehensive understanding of the relation between data distributions and models.

# 6 Related Work

In terms of the convergence trend of DNNs, existing works focus on the convergence of single models throughout the training process. The parameters of DNNs have been demonstrated to converge to global minima throughout the training progress (Goodfellow et al., 2014b; Li et al., 2018; Allen-Zhu et al., 2019; Liu et al., 2020; Damian et al., 2021; Refinetti et al., 2023; Suh and Cheng, 2024). Recent years, the studies of benign overfitting also suggest that increasing model capacities can improve the performance instead of exacerbating the overfitting issue (Bartlett et al., 2020; Nakkiran et al., 2021; Cao et al., 2022). While the studies of input gradients span into an abundant but extremely complicated spectrum. Among them, the area that is the most related to our work is the XAI domain, where the input gradient and its variants are crucial in revealing the models' internal mechanisms (Simonyan et al., 2013; Springenberg et al., 2014; Selvaraju et al., 2017; Sundararajan et al., 2017; Adebayo et al., 2018; Shah et al., 2021). On the other hand, the studies of the distribution of optimized models have received little attention. Such topics are slightly dipped in the efforts to demystify the source of capability of deep ensembles (Lee et al., 2015; Fort et al., 2019; Allen-Zhu and Li, 2020; Kobayashi et al., 2021; Abe et al., 2022; Ganaie et al., 2022; Theisen et al., 2023) and their implications (Lakshminarayanan et al., 2017; Geiger et al., 2020; Yang et al., 2021; Chen et al., 2023). Thus, to our knowledge, the studies on the distribution of optimized models remain a novel topic.

## Acknowledgement

This work was supported, in part, by the Defense Advance Research Projects Agency (Prime contract award number: HR0011222003, subcontract award number: 2103299-01, grant number: 13001129), the EMBRIO Institute, contract #2120200, a National Science Foundation (NSF) Biology Integration Institute, and NSF IIS #1955890, IIS #2146091, IIS #2345235. The content of the information does not necessarily reflect the position of the US Government. No official endorsement should be inferred. Approved for public release; distribution is unlimited.

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

# A  Additional Results of the Similarities

**The illustration of CIFAR-100 & TinyImagenet.** In addition to the cosine similarity $\rho_{ind}(k_1, k_2)$ shown in Figure 2, we include more sophisticated datasets such as CIFAR-100 and TinyImagenet in Figure 12. It is observed that the observed phenomena, where $\rho_{ind}(k_1, k_2)$ tend to increase w.r.t. both $k_1, k_2$, hold across different datasets. It is also worth noticing that compared with the results of CIFAR-10 shown in Figure 2, the resulting $\rho_{ind}(k_1, k_2)$ of CIFAR-100 and TinyImagnet (Figure 12) shows another peak at around $k \approx 10$ (see Figure 12(e)). This is related to the deep double descent phenomenon (Nakkiran et al., 2021), where when the complexities of the model and the dataset are comparable, the overfitting issue is at peak. For smaller such as CIFAR-10 or larger models such as ResNet, this issue becomes much less significant, as a very small $k$ value is already sufficient for the data distribution. Also, as the complexity of the data distribution increases, the cosine similarity decreases inevitably for the same model complexity. This suggests a less concentrated distribution of the optimized models of the same capacity compared with less complex dataset. Correspondingly, the results of the expectation over the model sets for $k \in K' = [10, 20, 40, 80, 160]$ are shown in Figure 13.

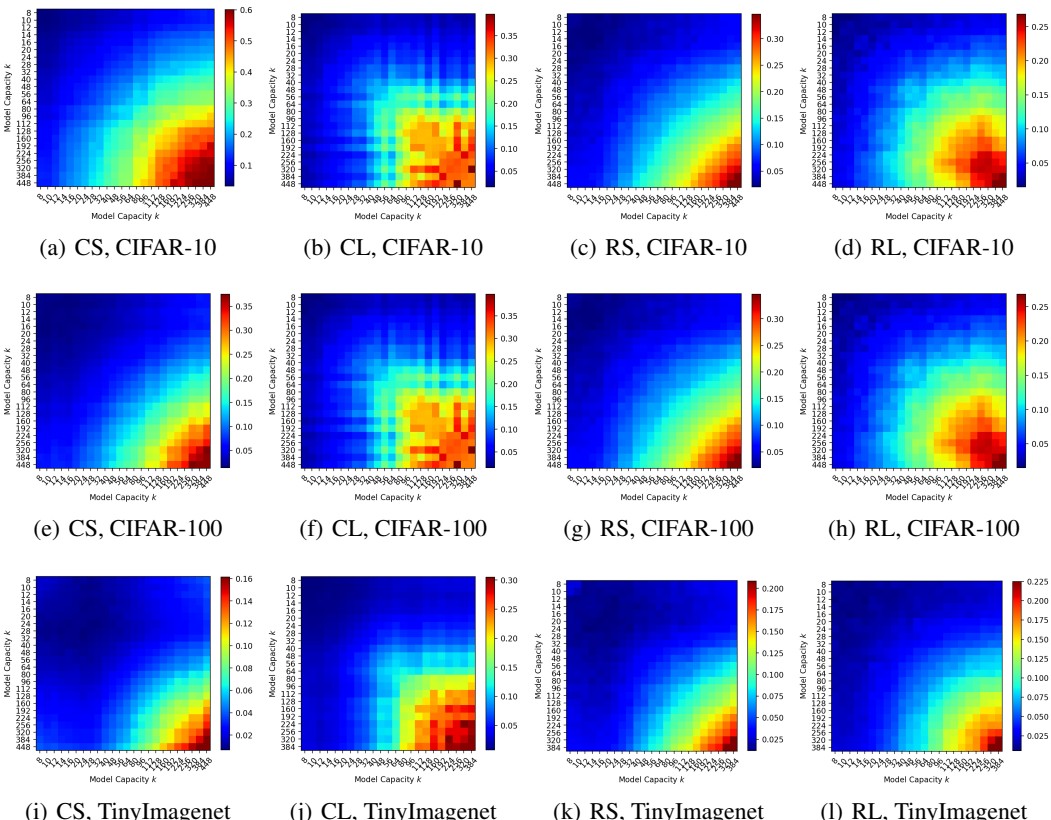

(a) CS, CIFAR-10  (b) CL, CIFAR-10  (c) RS, CIFAR-10  (d) RL, CIFAR-10

(e) CS, CIFAR-100  (f) CL, CIFAR-100  (g) RS, CIFAR-100  (h) RL, CIFAR-100

(i) CS, TinyImagenet  (j) CL, TinyImagenet  (k) RS, TinyImagenet  (l) RL, TinyImagenet

Figure 12: The expected similarity $\rho(k_1, k_2)$ between models of varying widths $k_1, k_2$. Here we include CNNSmall, CNNLarge, ResNetSmall, and ResNetLarge as $\mathcal{F}$. The values of $k_1, k_2$ determine the widths in each layer. Here the datasets are CIFAR-10 (top), CIFAR-100 (middle) and tinyImagenet (bottom).

**The Ablation of the Training Process.** To verify that the observed increasing trends of $\rho_{ind}(k_1, k_2)$ with model capacities are caused by the training process of DNNs instead of some normalization issue, we compare the similarity for models with initialized parameters. The results are shown in Figure 14. All three datasets and four model families are included. It can be clearly observed that, when the model parameters are initialized, the similarity between input saliency maps of different models are distributed randomly. The cosine similarity values are very concentrated around 0, which is the mean of random distribution. This verifies that the aforementioned increasing trends are caused by the optimization of models instead of normalization process.

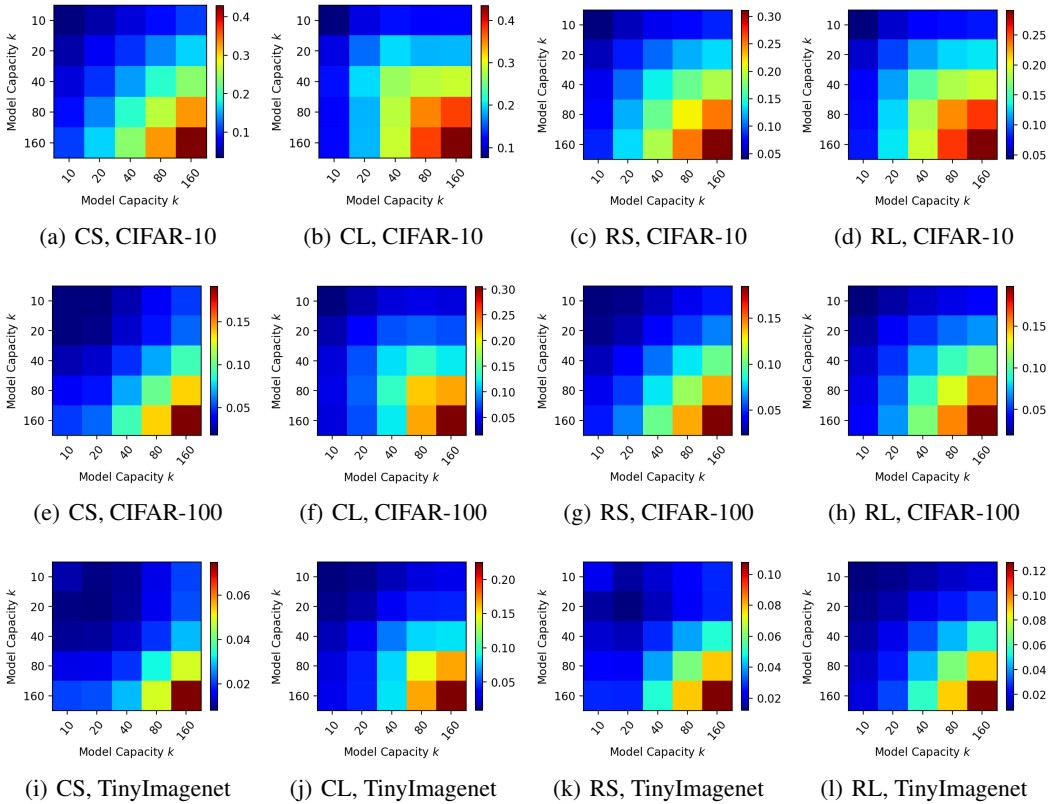

Figure 13: The expected similarity $\rho(k_1, k_2)$ between models of varying widths $k_1, k_2$. Here we include CNNSmall, CNNLarge, ResNetSmall, and ResNetLarge as $\mathcal{F}$. The values of $k_1, k_2 \in [10, 20, 40, 80, 160]$ determine the widths in each layer. Here the datasets are CIFAR-10 (top), CIFAR-100 (middle), and tinyImagenet (bottom).

**Softmax Activations.** Apart from the normalization concern, recent work in (Wang and Wang, 2022) demonstrate the difference between the input salience of the predicted logits and probabilities. As a result, we clarify that, although we define $f$ as the predicted logit (before softmax activations), this choice does not affect the observed increasing trend, no matter when the input salience is generated concerning the logit, probability, or the loss. The results of $\rho_{ind}(k_1, k_2)$, generated from the saliency maps w.r.t. the predicted probability (after softmax activations), are shown in Figure 15. It can be found that $\rho_{ind}(k_1, k_2)$ still increases with both $k_1$ and $k_2$.

# B Experiment Details

**Model Details** Throughout the experiments, we use CNNSmall, CNNLarge, ResNetSmall, and ResNetLarge as model families. Within each family, model width is controlled by the parameter $k$. And the model depths are controlled by the "Small" vs. "Large" suffixes. For CNNs, CNNSmall consists of convolutional layers with channels $[k, 2k, 4k, 8k]$, while CNNLarge repeats each layer twice: $[k, k, 2k, 2k, 4k, 4k, 8k, 8k]$. The details of CNNs are shown in Table 2. Additionally, for TinyImagenet, since the input data is of size $64 \times 64$, we increase the stride of the second `MaxPool2d` layer (Layer 10) to $4$. As for ResNets, we modify the width of ResNet-10 for ResNetSmall and ResNet-18 for ResNetLarge. Note that $k = 64$ ResNetSmall corresponds to ResNet-10, while $k = 64$ ResNetLarge corresponds to ResNet-18. The sizes of models are illustrated in Figure 16 as the # of trainable parameters.

It should also be noted that, ideally, CNNSmall and CNNLarge, ResNetSmall and ResNetLarge are considered as the same families due to the same architecture. However, since widths can be adjusted independently of the depth, while the adjustment of depth inevitable affects the width, we split them.

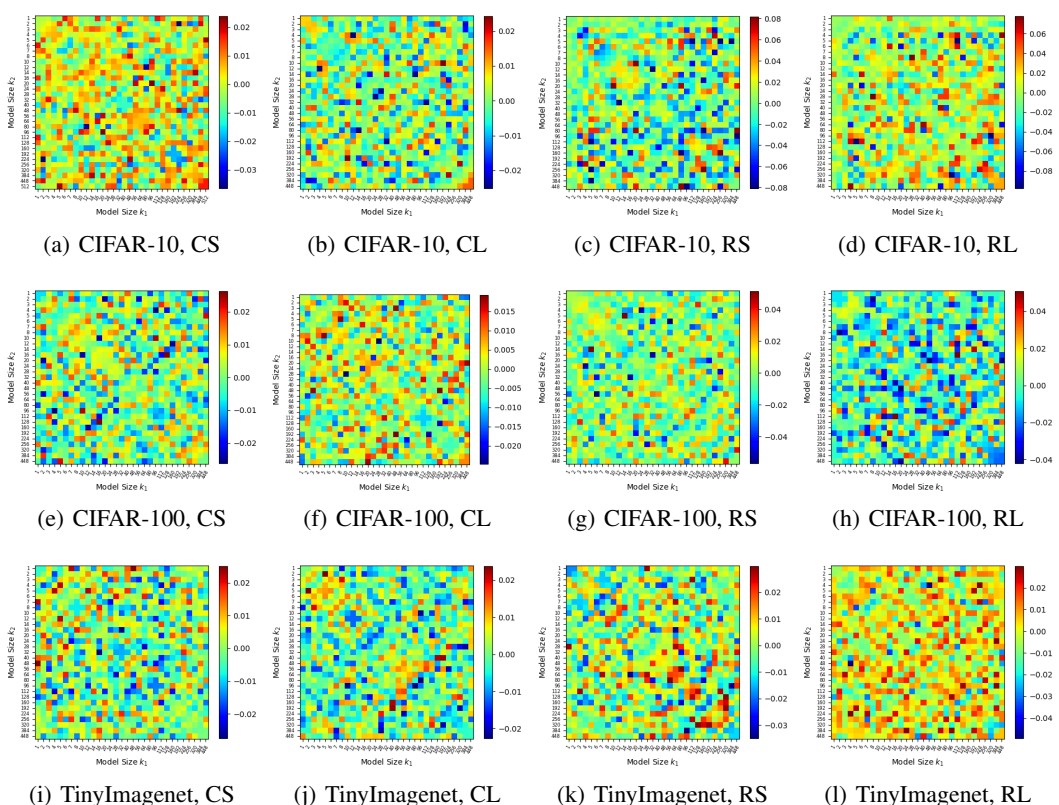

| | | | | |
|---|---|---|---|---|
| (a) CIFAR-10, CS | (b) CIFAR-10, CL | (c) CIFAR-10, RS | (d) CIFAR-10, RL |
| (e) CIFAR-100, CS | (f) CIFAR-100, CL | (g) CIFAR-100, RS | (h) CIFAR-100, RL |
| (i) TinyImagenet, CS | (j) TinyImagenet, CL | (k) TinyImagenet, RS | (l) TinyImagenet, RL |

Figure 14: The expected similarity $\rho(k_1, k_2)$ between models of varying widths $k_1, k_2$. Here we include CNNSmall (CS), CNNLarge (CL), ResNetSmall (RS), and ResNetLarge (RL) as $\mathcal{F}$. The values of $k_1, k_2$ determine the widths in each layer. All the models are initialized to random values without any optimizations. Here the datasets are CIFAR-10 (top), CIFAR-100 (middle), and TinyImagenet (bottom).

Table 1: The average Wasserstein distance $\mathbb{E}_{f^{(1)}, f^{(2)} \in \mathcal{F}(k)} \mathrm{WD}(\mathcal{S}(f^{(1)}), \mathcal{S}(f^{(2)}))$ with the standard deivation for all model families $\{CS, CL, RS, RL\} \times \{10, 20, 40, 80, 160\}$ over CIFAR-10/100 and TinyImagenet datasets. Note that here the baseline should be 2 since the cosine similarity lies in $[-1, 1]$. It is observed that deeper models usually have larger distances (CS vs. CL, RS vs. RL). We deduce that this is because of the training for deeper models is more difficult.

| CIFAR-10 | $k = 10$ | $k = 20$ | $k = 40$ | $k = 80$ | $k = 160$ |
|---|---|---|---|---|---|
| CS | $0.0081 \pm 0.0051$ | $0.0100 \pm 0.0060$ | $0.0106 \pm 0.0062$ | $0.0135 \pm 0.0094$ | $0.0153 \pm 0.0106$ |
| CL | $0.0143 \pm 0.0091$ | $0.0171 \pm 0.0144$ | $0.0322 \pm 0.0295$ | $0.0334 \pm 0.0287$ | $0.0345 \pm 0.0271$ |
| RS | $0.0097 \pm 0.0059$ | $0.0092 \pm 0.0055$ | $0.0088 \pm 0.0052$ | $0.0073 \pm 0.0043$ | $0.0061 \pm 0.0036$ |
| RL | $0.0096 \pm 0.0054$ | $0.0107 \pm 0.0065$ | $0.0090 \pm 0.0053$ | $0.0090 \pm 0.0057$ | $0.0163 \pm 0.0121$ |
| CIFAR-100 | $k = 10$ | $k = 20$ | $k = 40$ | $k = 80$ | $k = 160$ |
| CS | $0.0078 \pm 0.0052$ | $0.0056 \pm 0.0034$ | $0.0054 \pm 0.0031$ | $0.0060 \pm 0.0034$ | $0.0085 \pm 0.0056$ |
| CL | $0.0098 \pm 0.0059$ | $0.0099 \pm 0.0060$ | $0.0156 \pm 0.0106$ | $0.0184 \pm 0.0130$ | $0.0222 \pm 0.0146$ |
| RS | $0.0071 \pm 0.0034$ | $0.0066 \pm 0.0029$ | $0.0062 \pm 0.0030$ | $0.0061 \pm 0.0029$ | $0.0055 \pm 0.0029$ |
| RL | $0.0087 \pm 0.0042$ | $0.0079 \pm 0.0037$ | $0.0078 \pm 0.0037$ | $0.0065 \pm 0.0029$ | $0.0078 \pm 0.0044$ |
| TinyImagenet | $k = 10$ | $k = 20$ | $k = 40$ | $k = 80$ | $k = 160$ |
| CS | $0.0033 \pm 0.0018$ | $0.0021 \pm 0.0010$ | $0.0018 \pm 0.0007$ | $0.0028 \pm 0.0013$ | $0.0060 \pm 0.0049$ |
| CL | $0.0032 \pm 0.0016$ | $0.0032 \pm 0.0014$ | $0.0055 \pm 0.0029$ | $0.0129 \pm 0.0083$ | $0.0204 \pm 0.0155$ |
| RS | $0.0083 \pm 0.0052$ | $0.0058 \pm 0.0038$ | $0.0049 \pm 0.0025$ | $0.0045 \pm 0.0023$ | $0.0041 \pm 0.0018$ |
| RL | $0.0062 \pm 0.0040$ | $0.0060 \pm 0.0035$ | $0.0055 \pm 0.0030$ | $0.0055 \pm 0.0031$ | $0.0057 \pm 0.0031$ |

But our experiments in Section 3.1 verify that the depths do not affect the population mean of model

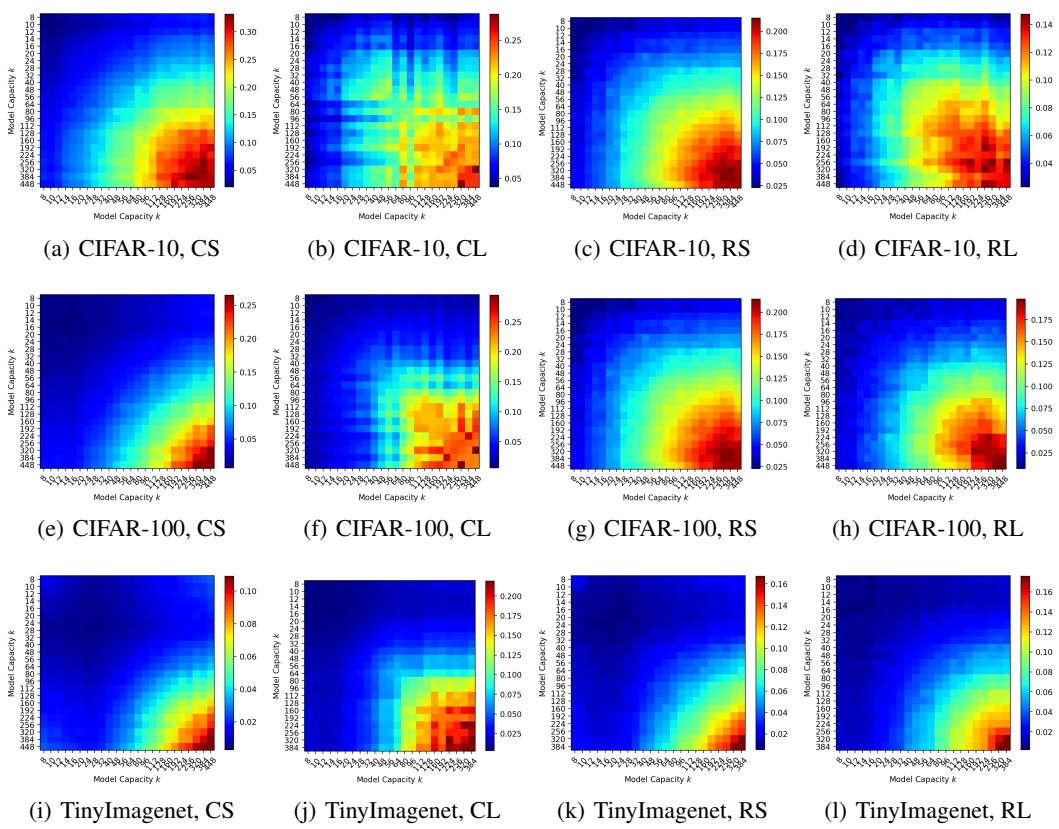

(a) CIFAR-10, CS    (b) CIFAR-10, CL    (c) CIFAR-10, RS    (d) CIFAR-10, RL

(e) CIFAR-100, CS    (f) CIFAR-100, CL    (g) CIFAR-100, RS    (h) CIFAR-100, RL

(i) TinyImagenet, CS    (j) TinyImagenet, CL    (k) TinyImagenet, RS    (l) TinyImagenet, RL

Figure 15: The expected similarity $\rho(k_1, k_2)$ between models of varying widths $k_1, k_2$. Here we include CNNSmall, CNNLarge, ResNetSmall, and ResNetLarge as $\mathcal{F}$. The values of $k_1, k_2$ determine the widths in each layer. In particular, all the cosine similarities are between the input saliency maps of the predicted probabilities instead of the predicted logits. Here the datasets are CIFAR-10 (top), CIFAR-100 (middle) and tinyImagenet (bottom).

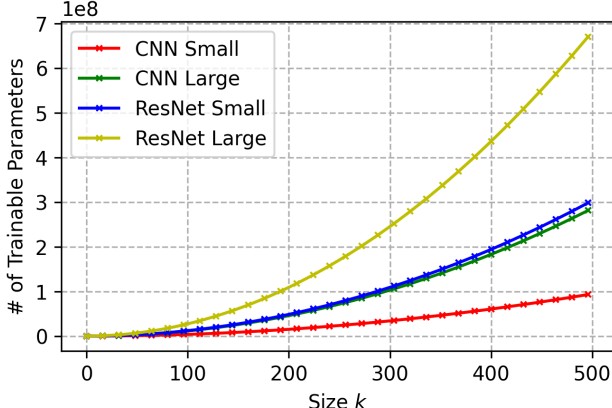

Figure 16: The # of trainable parameters of models vs. the width parameter $k$ for each architecture.

Table 2: CNNSmall Model Details

| Layer | Type | Parameters |
|---|---|---|
| 0 | Conv2d | 3 inch, $k$ outch, ks 3, stride 1, padding 1 |
| 1 | BatchNorm2d | - |
| 2 | ReLU | - |
| 3 | Conv2d | $k$ inch, $2k$ outch, ks 3, stride 1, padding 1 |
| 4 | BatchNorm2d | - |
| 5 | ReLU | - |
| 6 | MaxPool2d | ks 2, stride 2 |
| 7 | Conv2d | $2k$ inch, $4k$ outch, ks 3, stride 1, padding 1 |
| 8 | BatchNorm2d | - |
| 9 | ReLU | - |
| 10 | MaxPool2d | ks 2, stride 2 |
| 11 | Conv2d | $4k$ inch, $8k$ outch, ks 3, stride 1, padding 1 |
| 12 | BatchNorm2d | - |
| 13 | ReLU | - |
| 14 | MaxPool2d | ks 8, stride 8 |
| 15 | Flatten | - |
| fc | Linear | in_features=80, out_features n_class |

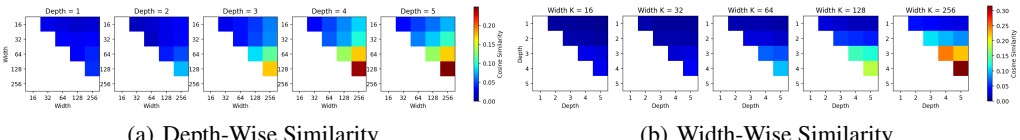

(a) Depth-Wise Similarity        (b) Width-Wise Similarity

Figure 17: The cosine similarities between CNN models with different widths (left) and depths (right). Widths and depths are enumerated in $\{16, 32, 64, 128, 256\}$ and $\{1, 2, 3, 4, 5\}$. Note that to scale the depths without affecting widths, here we fix the widths of all layers to k instead of $[k, 2k, 4k, 8k]$ as the manuscript

distributions. In experiments, we set $\mathcal{X}$ as the first 1000 samples of the unshuffled testing set of each dataset (CIFAR-10/100, TinyImagenet).

## C   Additional Settings

This work aims to reveal the convergence trend of the distribution of model behaviors under the stochasticity of the training criterion. This does not limit the conclusion to the specific criterion described above. Distinct training criteria can lead to different distributions of trained models. However, these different distributions of trained models all satisfy the revealed trend. To verify this, we present additional experiments to investigate possible variants such as (1) depths and widths; (2) learning rates; (3) batch sizes; (4) solvers (5) initializations, and (6) other model architectures. Note that as studied in Figure 6 the manuscript, $\rho_{ind}$ can be a computationally efficient compromise of $\rho$. Therefore, we studied $\rho_{ind}$ in these additional experiments. Besides, there exists enumerous possible combinations of different variants of different aspects. As a result, here we only vary these settings partially since enumerating the entire grid is infeasible. The results as detailed as follows.

**Depths and widths.** The scale of depths is not as straightforward as the width since modifying depths may change widths as well. Therefore, in the manuscript we study the influence of depth by setting -small and -large variations. Here we present additional results that study the influence of depths continuously, with 1-5 layers, each of which is followed by a max-pooling layer with stride 2. Finally, an adaptive pooling layer is appended at the end. To rule out the influence of widths (channels), all layers have the same width, determined by $k$. e.g., For the 4-layer scenario, the intermediate layers have widths $[k, k, k, k]$ instead of $[k, 2k, 4k, 8k]$ in the manuscript. The results are shown in Figure 17. It can be found that (1) Given a fixed depth or width, the influence of the

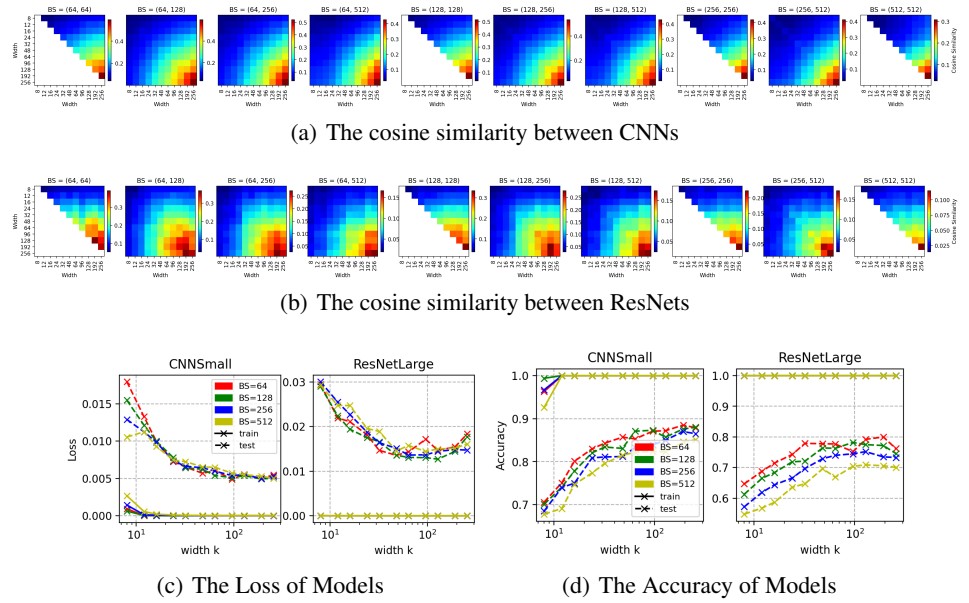

(a) The cosine similarity between CNNs

(b) The cosine similarity between ResNets

(c) The Loss of Models

(d) The Accuracy of Models

Figure 18: (a) and (b) illustrate the cosine similarity between (a) CNNs and (b) ResNets with different batch sizes in $\{64, 128, 256, 512\}$. (c) shows the loss and (d) shows the accuracy of trained models.

other factor is similar when scaled up. (2) Depths are slightly different from widths. Larger widths lead to higher similarities, while closer structures in depths have higher similarities. For widths (left), the similarity always increases left-to-right and top-to-bottom. But for depths (right), pairs near the diagonal have higher similarities.

**Batch Sizes.** We investigate the influence of batch sizes, varying in $\{64, 128, 256, 512\}$ The results are shown in Figure 18. It can be observed that although different batch sizes lead to different performance (e.g. testing accuracy), the convergence trend holds in all scenarios.

**Learning Rates.** We test different learning rates on how they affect the results. We include `1e-1`, `1e-2`, default, where "default" refers to the criterion used in the manuscript. As shown in Figure 19, the revealed trend is preserved in all learning rates. It is also worth noticing that learning rates affect ResNets more than CNNs.

**Solvers.** Apart from SGD, we include Adam, AdamW, and SGD w/ momentum. For Adam and AdamW we set the learning rate to 1e-3, while SGD w/ momentum uses a learning rate of 1e-1 with a momentum of 0.9. The results are shown in Figure 20. Although different solvers lead to models of different performances, they all preserve the same convergence trend.

Table 3: The comparison between the similarities between single models of different criteria.

|  | (a) Diff. Init.; Same Order | (b) Diff. Init.; Diff. Order | (c) Same Init. $\theta_0$; Diff Order | (d) Same Init. $\theta_1$; Diff Order |
|---|---|---|---|---|
| # of pairs | 100 | 4095 | 4095 | 4095 |
| mean of $\rho_{ind}$ | 0.0758 | 0.0753 | 0.0879 | 0.0855 |
| std. of $\rho_{int}$ | 0.0038 | 0.0037 | 0.0042 | 0.0048 |

**Initializations.** Given a training scheme and model family $\mathcal{F}(k)$, the training procedure leads to a distribution of trained models $p(f)$. When the initialization is fixed to $\theta$, the training procedure is essentially sampling from the conditional distribution $p(f|\theta)$ instead of the unconditional distribution $p(f)$. We then studied the difference between the unconditional distribution $p(f)$ and the conditional distribution $p(f|\theta_0)$. We focus on two conditional distributions $p(f|\theta_0)$ and $p(f|\theta_1)$, where $\theta_1$

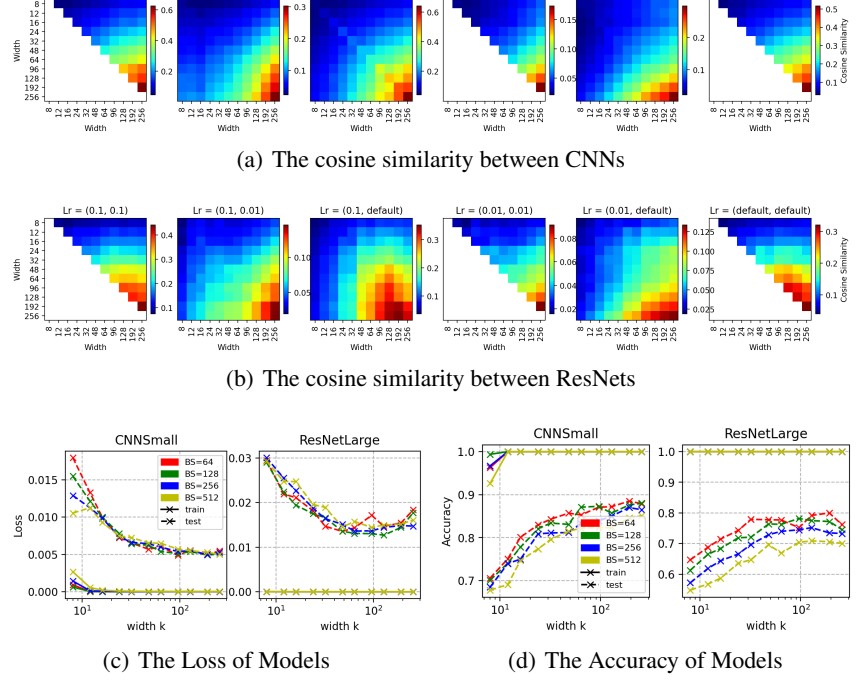

(a) The cosine similarity between CNNs

(b) The cosine similarity between ResNets

(c) The Loss of Models

(d) The Accuracy of Models

Figure 19: (a) and (b) illustrate the cosine similarity between (a) CNNs and (b) ResNets with different batch sizes in $\{64, 128, 256, 512\}$. (c) shows the loss and (d) shows the accuracy of trained models.

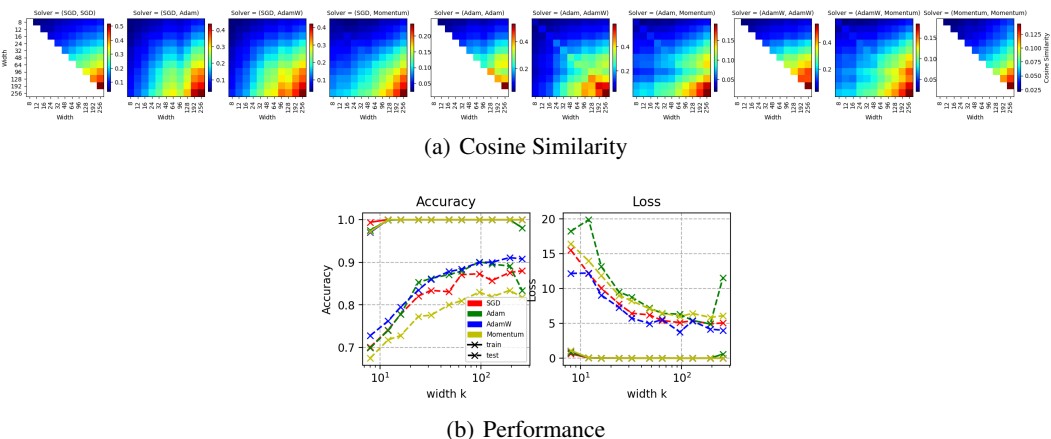

(a) Cosine Similarity

(b) Performance

Figure 20: (a) The cosine similarity between CNN models with different solvers in {SGD, Adam, AdamW, SGD w/ Momentum}. (b): The accuracy and loss of trained models.

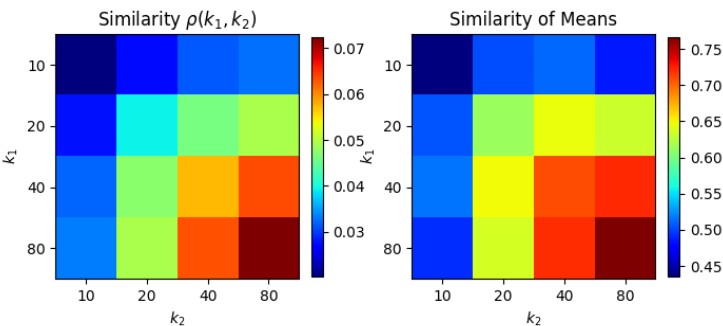

Figure 21: The cosine similarity between Vision Transformers (ViTs) on CIFAR-10. The capacity is controlled by $k \in \{10, 20, 40, 80\}$, where the embedding dimension is $4k$, separated to $k/2$ heads. The left subfigure shows the mean of the similarity. The right subfigure shows the similarity of the population mean.

represents the initializations under seed=1 and $\theta_0$ represents those under seed=0. Other settings are identical. We thus have $f_1^0, \cdots, f_{100}^0 \sim p(f|\theta_0)$ and $f_1^1, \cdots, f_{100}^1 \sim p(f|\theta_1)$. The superscript indicates the initialization seeds and the subscript indicates the training seeds.

First, we notice immediately that the training seeds for both $\theta_0$ and $\theta_1$ are 1 100. This means that $\forall i, f_i^0, f_i^1$ differ only in initializations. We inspect (a) $\rho_{ind}(f_i^0, f_i^1)$ (100 pairs) to see if they have exceptional similarity compared with (b) $\rho_{ind}(f_i^0, f_j^1), i \neq j$ ($\binom{100}{2} = 4950$ pairs). Besides, within the same condition, all models only differ in terms of the orders of the training batch. We thus also inspect the similarity of all models of the same condition: (c) $\rho_{ind}(f_i^0, f_j^0), i \neq j$ and (d) $\rho_{ind}(f_i^1, f_j^1), i \neq j$. Each of them has $\binom{100}{2} = 4950$ pairs.

As demonstrated in Table 3, (i) the comparison between (a) and (b) indicates that with different initializations, the same order of batches in the training procedure does not contribute to higher similarities. (ii) The comparison among (b)(c)(d) indicates that the same initialization indeed leads to higher similarity even though the order of batches is distinct. It should be noted that the contributions of batch orders and initializations are also affected by the number of epochs. Intuitively, more training epochs should lead to smaller contributions from the initializations but greater contributions from the batch orders.

**Vision Transformers.** Vision Transformers (ViT) have risen in recent years as another powerful architecture for CV tasks. Here we include a brief study of ViTs to demonstrate that the discovered phenomenon holds across different architectures.

Specifically, we train vision transformer (ViT) models on CIFAR-10 with varying capacities controlled by $k \in \{10, 20, 40, 80\}$. CIFAR-10 has an input size of $32 \times 32$ pixels, thus the patch size is set as $4 \times 4$, resulting in $8 \times 8$ patches. The embedding size is set to $4k$, divided by $k/2$ heads, and we set the depth to 8. The seeds vary in 1-100 and results in 100 trained models of each $k$. We study the mean of the similarity $\rho(k_1, k_2)$ (i.e. the same experiments as Figure 3 in the manuscript) and the similarity of the population mean (i.e. the same as Figure 5 in the manuscript). The results are shown in Figure 21. It can be observed that although distinct from convolutional layers, the transformer structure also has the discovered convergence trend. It can also be noted that the degree of dispersion of ViTs is much higher than CNN-based models.

In conclusion, although training schemes can affect the resulting distributions of models, the influence of the model capacity stays invariant across different criteria.

# D   Uniform Distributions on the Hypersphere

According to (Muller, 1959), due to the spherical symmetry property of the zero-mean Gaussian distribution, the cosine similarity between two Gaussian variables are actually uniformly distributed

over the hypersphere $\mathcal{S}^{d-1}$ in $\mathbb{R}^d$.[3] Therefore, the cosine similarity between two i.i.d. multivariate Gaussian tensors is essentially the inner product between two i.i.d. uniform tensors over the hypersphere. Formally, suppose $X, Y$ to be high-dimensional i.i.d. random variables of dimension $d$ that follow the Gaussian distribution $\mathcal{N}(\mathbf{0}, I_{d \times d})$. The cosine similarity is $Z = \frac{X^T Y}{\|X\| \cdot \|Y\|}$. WLOG, it suffices to consider the scenario where $\frac{Y}{\|Y\|} = \boldsymbol{e}_1 = (1, 0, \cdots, 0)$. And it is written as

$Z = \boldsymbol{e}_i^T \frac{X}{\|X\|} = \frac{X_1}{\|X\|} = \frac{X_1}{\sqrt{\sum_{i=1}^d X_i^2}} = \sqrt{\frac{X_1^2}{X_1^2 \sum_{i=1}^d X_i^2}}$. Note that $X_1^2 \sim \chi^2(1), \sum_{i=2}^d X_i^2 \sim \chi^2(d-1)$. As a result, $Z^2 = \frac{X_1^2}{X_1^2 + \sum_{i=2}^d X_i^2}$ follows a beta distribution $Z^2 \sim Beta(\frac{1}{2}, \frac{d-1}{2})$. The pdf is thus $f_{Z^2}(x) = \frac{x^{-1/2}(1-x)^{(d-3)/2}}{B(1/2, (d-1)/2)}$. And then when $Z > 0$,

$$f_Z(x) = f_{Z^2}(x^2)|2x| \tag{7}$$

$$= \frac{(x^2)^{-1/2}(1 - x^2)^{(d-3)/2}}{B(1/2, (d-1)/2)}|2x| \tag{8}$$

$$= \frac{2}{B(\frac{1}{2}, \frac{d-1}{2})}(1 - x^2)^{(d-3)/2} \tag{9}$$

According to (Smith et al., 2023), let $u = (1+x)/2$, then $x^2 = (2u-1)^2$. Then this can be simplified to

$$f_U(u) = \frac{2}{B(\frac{1}{2}, \frac{d-1}{2})}(1 - (2u-1)^2)^{(d-3)/2} \cdot \frac{\mathrm{d}x}{\mathrm{d}u} \tag{10}$$

$$= \frac{1}{B(\frac{1}{2}, \frac{d-1}{2})}2^{d-2}u^{(d-1)/2-1}(1 - u)^{(d-1)/2-1} \tag{11}$$

$$= \frac{1}{B(\frac{1}{2}, \frac{d-1}{2})2^{2-d}}u^{(d-1)/2-1}(1 - u)^{(d-1)/2-1} \tag{12}$$

$$= Beta(\frac{d-1}{2}, \frac{d-1}{2}) \tag{13}$$

This is because

$$B(\frac{d-1}{2}, \frac{d-1}{2}) = B(\frac{d-1}{2}, \frac{d+1}{2}) \cdot 2 \tag{14}$$

$$= \frac{\Gamma(\frac{d-1}{2})\Gamma(\frac{d+1}{2})}{\Gamma(d)} \cdot 2 \tag{15}$$

$$= \frac{\Gamma(\frac{d-1}{2})\Gamma(\frac{d+1}{2})}{2^{d-1}\pi^{-1/2}\Gamma(\frac{d-1}{2})\Gamma(\frac{d+1}{2})} \cdot 2 \tag{16}$$

$$= 2^{2-d}\frac{\Gamma(\frac{d-1}{2})\Gamma(\frac{1}{2})}{\Gamma(\frac{d}{2})} = B(\frac{1}{2}, \frac{d-1}{2})2^{2-d} \tag{17}$$

Finally, we are able to conclude that $\frac{1+Z}{2} \sim Beta(\frac{d-1}{2}, \frac{d-1}{2})$, where $Z$ is the cosine similarity between two i.i.d. $d$-dimensional Gaussian vectors. This result suggests that the distribution of $Z$ will become very concentrated around 0. And this concentration exacerbates exponentially with the dimension $d$. Given a cosine similarity level $\rho$, the probability $\mathbb{P}(Z > \rho)$ can be extremely small, and also deceases exponentially with $\rho$, too. We visualize the relation between the probability $\mathbb{P}(Z > \rho)$ and $\rho$ with varying dimensions $d$ in Figure 22. In the low-dimensional space such as $d = 3$, the distribution of the cosine similarity is close to uniform as humans' intuition. However, as the dimension increases, the cosine similarity is very unlikely to maintain high, as demonstrated by the curves of $d = 3 \times 8 \times 8 = 48$ (orange) and $d = 3 \times 32 \times 32 = 3072$ (green).

---

[3]This is because of the rotation-invariance. Let $O \in \mathbb{R}^{d \times d}$ be any orthonormal matrix, then after the rotation we obtain $\frac{OX}{\|X\|} = \frac{OX}{\|OX\|}$. Since $OX \sim \mathcal{N}(\mathbf{0}, I)$, too, we know that $\frac{OX}{\|X\|}$ and $\frac{X}{\|X\|}$ are from the identical distribution.

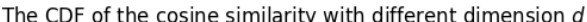

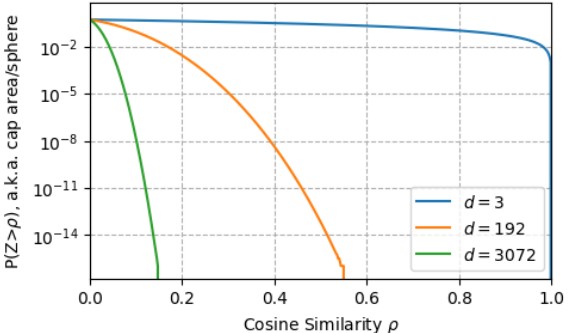

Figure 22: The relation between the probability $\mathbb{P}(Z > \rho)$ and the cosine similarity value $\rho$.

## E   Supplementary Experiment Results

Note that all the experiments are carried across three datasets CIFAR-10/100 and TinyImagenet, along with four model families CNNSmall, CNNLarge, ResNetSmall, and ResNetLarge. Due to the space limit, only part of the selected results can be put in the manuscript. Therefore, we defer the results with different models/datasets into this section for the audience' reference. All the conclusions drawn from the experiment results shown in the manuscript hold for the results demonstrated here.

In Figure 23, we present the complementary results of Figure 6 on CIFAR-100 and TinyImagenet. It can be clearly observed that $\rho_{ind}$ and $\rho$ have very similar values, after taking the expectation over $\mathcal{X}$.

Figure 25 shows the results of black-box attack results between different models. All other settings are identical to the results shown in Figure 10, but with CIFAR-100 and TinyImagenet instead of CIFAR-10. It is observed that the capacity of models has significant influence to the models' robustness and black-box attack transferability. And this trend is highly correlated to the similarity $\rho_{ind}, \rho$ as demonstrated in Figures 2, 3, 12, 13 and 15.

Similarly, Figure 26 shows supplementary results of Figure 11. All settings are identical except for the datasets. CIFAR-100 and TinyImagenet are tested instead of CIFAR-10. It can be observed that using the estimated mean gradient direction (blue), the performance drops much more significantly than the attack from single models of either the exact same family as the target model (green) or the largest single model tested ($k = 448$, red). Note that due to the complexity of TinyImagenet, in the overfitting phase (i.e. when the target models' capacities are comparable to the dataset (Nakkiran et al., 2021)), the single-model black-box attack results in opposite effect – the prediction actually increases.

In Figure 24, we present CIFAR-100 and TinyImagenet results as supplementary of Figure 9. It can be observed that the testing loss is highly correlated to the expectation of $t = \boldsymbol{u}^T \boldsymbol{\mu}(\boldsymbol{x})$. Such phenomena are also consistent across different model families and datasets. For both single models and ensembles, the closer they are to the convergent limiting point (i.e. larger $\mathbb{E}[t]$), the higher testing performance they have. Note that here for the sake of consistency, we approximate $\boldsymbol{\mu}(\boldsymbol{x})$ through $\tilde{\boldsymbol{\mu}}(160; \boldsymbol{x})$. Therefore, in the ensemble experiments (left of each subfigure), $k = 160$ is omitted.

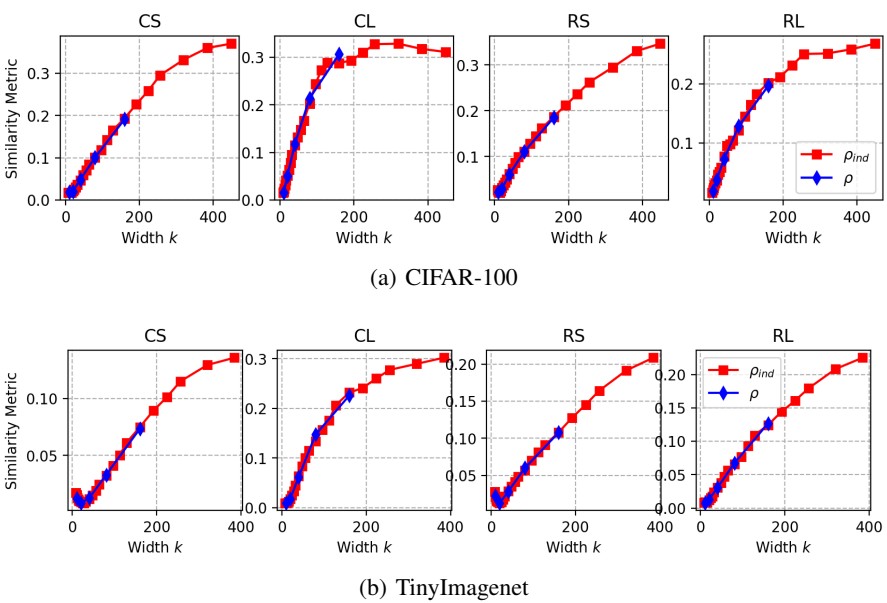

Figure 23: Illustration of (red) $\rho_{ind}(f^{(1)}, f^{(2)})$, $f^{(1)}, f^{(2)} \in \mathcal{F}(k)$ and (blue) $\rho(k, k)$ on (a) CIFAR-100 and (b) TinyImagenet.

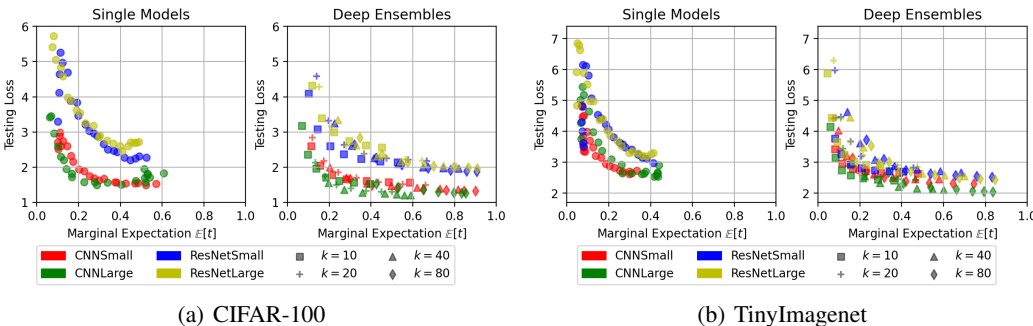

Figure 24: The illustration of the relation between the expected testing loss $\mathbb{E}_{\mathcal{X}}[\mathcal{L}]$ and the marginal expectation $\mathbb{E}_{\mathcal{X}}[t]$. Both (a) CIFAR-100 and (b) TinyImagenet results are shown as supplementary to Figure 9. Models are from (i) single models with varying structure; and (ii) deep ensembles with varying members. Each color represents a model family.

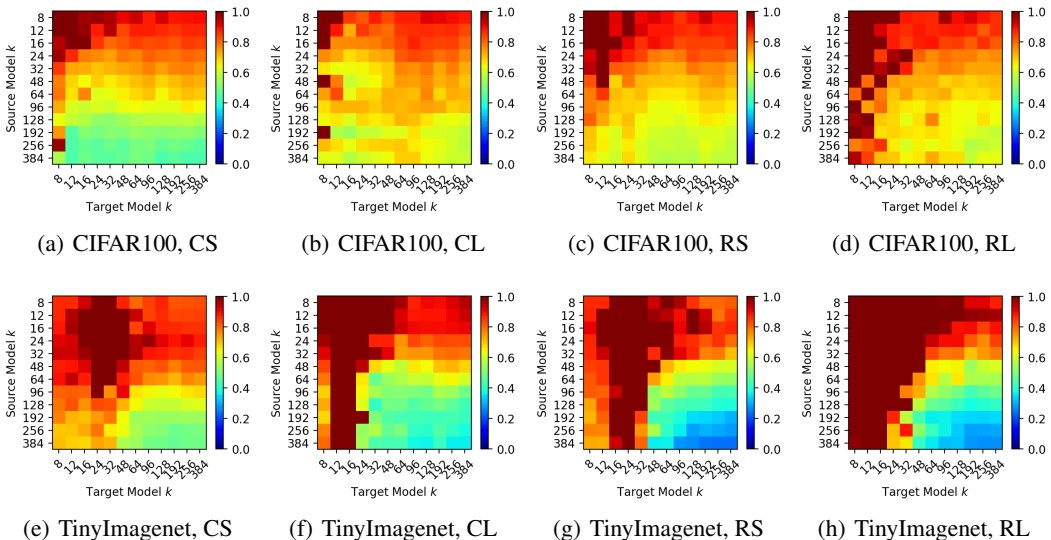

Figure 25: The results of single model black-box attack. The value of each entry is $\alpha(f^{(1)}, f^{(2)})$, $f^{(1)} \in \mathcal{F}(k_1), f^{(2)} \in \mathcal{F}(k_2)$ for different model capacities. Here $k_1$ is the width parameter of the source model and $k_2$ is the width parameter of the target model.

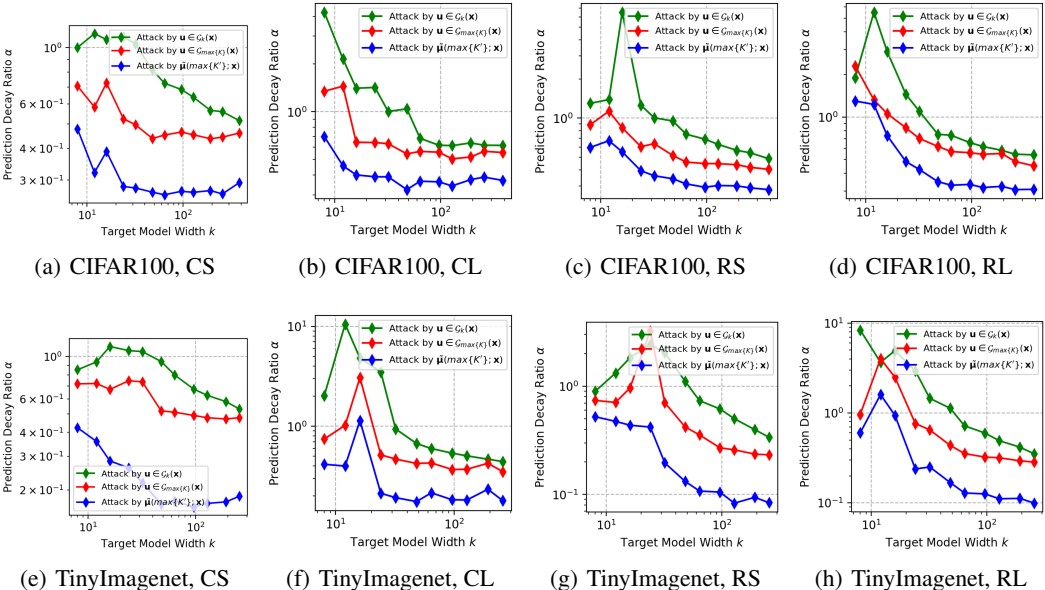

Figure 26: The comparison between the single-model attack from the largest model (red), the single-model attack from the very same structure (green), and the attack by the mean direction (blue). The top row shows the results of CIFAR-100, and the bottom row shows the results of TinyImagenet. Some figures' $y$-axis are set to logarithm for clarity.

