# OpenReview forum: "Great Minds Think Alike: The Universal Convergence Trend of Input Salience"
_NeurIPS.cc/2024/Conference — NeurIPS 2024 poster_

### Official Review · Reviewer_We4a · 2024-07-12

**Soundness:** 3
**Presentation:** 2
**Contribution:** 2
**Rating:** 4
**Confidence:** 4

**Summary:**

This paper studies the resemblance of the functions obtained by training neural networks across various network depth and width. They do so by looking at the gradient field of the learned functions $f_1, f_2$, and defining their similarity by taking the cosine similarity between the gradients, then taking independent average over the inputs of the functions. They empirically observe that increasing the width results in higher similarity between the gradient fields, from which the deduce to properties: (1) the average gradient norm increase with width of the net, and (2) the direction of the average almost does not change when going to networks with larger width, but the variance decrease and the distribution becomes more concentrated. They also calculate the similarity measure and some properties of the gradient distribution assuming the distribution is gaussian or spherically symmetric. The major claim of. this work is summarized in hypothesis H1 and H2 on lines 145 and 160.

**Strengths:**

Authors formulate an interesting hypothesis based on empirical observation regarding the behavior of the gradient field of learned functions in DNNs: the direction of the vector does not change too much when we increase the size of the network. They form this hypothesis from the observation that the average dot product of the gradient fields across networks with different width increases compared to the norm of the average of the gradient field for the smaller network.

**Weaknesses:**

My major concerns:
(1) The claims are not backed my sufficient reasoning and relevant experiments. Note that the similarity measure is a function of the dot product of the average normalized gradients, so seeing an increasing trend on this quantity can be the result of two things: (*) the angle between the average of the normalized gradients doesn't change or gets smaller, or (**) the size of the average itself is increasing. Note that either of (*) or (**) is sufficient to happen to see an increasing value of cosine similarity that the authors have defined. However, observing an empirical increase in the value of similarity by going to larger nets, the authors conclude both of (*) and (**) at the same time, while it might be the case that (**) happens, ie.e. the size of the average is increasing fast, so that even though the direction of the averages are getting further apart, the overal dot product is still increasing. Therefore, in order to scientifically claim that both (*) , (**), one approach is that authors study the angle between the normalized averages (or dot product of NORMALIZED averages) separately. Specially because these claims are not backed by theoretical arguements.

(2) The authors discuss the 'saw distribution' and modeling the behaviour of gradients with this dsitribution assuming the actuall gradient distribution is spherically symmetric (or close to it), and assumption that seems very strong and probably incorrect for actual neural nets. Is there a particular observation for this assumption? Authors discuss some high level calculation with this distribution on line 246, but I didn't find any calculation in appendix.

(3) the writing is vague and it does not flow. In particular, the story for the theoretical calculation with gaussian distribution is not explained, and the assumptions are not justified at all. The section regarding saw distribution needs a lot of changes to become mroe clear.

Some minor points:
- the authors claim that the fact that given networks A, B where A has larger width, they claim that the observation that the average similarity of A to B is larger than that of B to itself is 'counter-intuitive' and 'paradoxical'. This is a wrong claim and needs to be omited from the paper; the reason that it is not paradoxical is you are working with 'some measure of similarity' rather than 'distance' or 'metric', for a simple illustrative example, if you define similarity of two numbers a, b to be their product ab, then if a > b > 0, ab > b^2 but it doesn't mean that because a, b are more similar, we should conclude that a is 'closer' to b than b to itself.

- In Equation (5) you are estimating the normalized average of the gradient which is used in calculating your similarity measure, but I think the quantity of interest is the normalized average of the normalized gradients (because of the definition of cosine similarity that you use)?

minor comments:
- at H1 hypothesis in line 145:  the mean vector \mu is not defined at that point of the paper
-typo in line 158 -> should be k_1 and k_2

**Questions:**

What is the conclusion for training/optimization/generalization efficiency the authors look into obtain from this observation? To me the average of normalized gradient seems not to be correlated with generalization ability of neural nets. maybe the authors can add more concluding points or remarks about their empirical observation.

- What is the difference between the message of Figure 3 and 5?

- Are you assuming the gradient distribution is gaussian in section C?

**Limitations:**

The claim needs further be backed by more delicate experiments. Furthermore, the assumptions for calculation of the behavior of the similarity measure using certain classes of distributions needs to be justified at least in some regime of training DNNs. For the rest see weaknesses above.

---

> ### Author Rebuttal · Authors · 2024-08-07
>
> ## Response to Reviewer We4a
> We thank the reviewer for acknowledging our contribution. Regarding the reviewer's concerns, we clarify that there is some communication in the empirical verifications.
>
> First, we would like to answer the clarification questions:
>
> **[Eq. 5 (W4)]**
>
> We thank the reviewer for pointing out this typo! As the empirical estimation of $\mu$, $\tilde{\mu}$ is computed as the **normalized average of the normalized gradients** throughout the experiments. This is the same as how $\mu$ is defined in L152 (where $u$ is the normalized gradient). It is defined as
> $$\tilde{\mu}(k;x)=\frac{1}{M}\sum_{i=1}^M\frac{\nabla_x f_i(x)}{||\nabla_x f_i(x)||})/||\frac{1}{M}\sum_{i=1}^M\frac{\nabla_x f_i(x)}{||\nabla_x f_i(x)||}||\approx \mu(k;x),f_i\in\mathcal{F}(k).$$
> We will revise it in the manuscript.
>
> **[Fig. 3 and 5 (Q1)]**
>
> - Fig. 3 presents the results of $\rho(k_1,k_2)$, which is **the average of the cosine similarity between two families**. This is defined in L110-112 and serves as a starting point for the two hypotheses. The increase in the diagonal motivates and verifies H1. The increase in the rows and columns motivates and (partially) verifies H2 as mentioned by the reviewer.
> - Fig. 5 shows the results of **the cosine similarity between the averages of the normalized gradients of two families**. This is the angle between the population means of two $\mathcal{G}$s. As also mentioned by the reviewer, this verifies H2.
>
> **[Gradient Similarities (W1)]**
>
> After clarifying the miscommunication, we now address the reviewer's major concerns:
>
> The reviewer's observation of the two possible factors is insightful. Both of them are already considered **separately** in the manuscript. (ii) is discussed in the intra-family hypothesis.
> As for (i), there were some miscommunications due to the typos in Eq. 5. The **angles between the average of the normalized gradients** are exactly what we used to verify H2. The results are presented in Fig. 5, with definitions in the caption as $\mathbb{E}\_{x\in\mathcal{X}}CosSim(\tilde{\mu}\_j(x;\mathcal{F}),\tilde{\mu}_{j'}(x;\mathcal{F}))$. This is the dot product of normalized averages of the normalized gradients suggested by the reviewer. The results demonstrate an extremely high cosine similarity between the means and thus provide verifications to H2.
>
> Furthermore, we reassure the reviewer with the results in Fig. 10. The mean attack is performed by the means estimated using the family of $k=160$ (L291-297). Thus if the concerned scenario is true, the mean attack would not lead to the best results when attacking other families.
>
> **[Saw Distribution (W2,W3)]**
>
> - The symmetry assumptions (W2). To study the degree of concentration of the gradients, we study the cosine similarity $t = u^T\mu$ between an individual gradient and the mean. This naturally leads to rotationally symmetric distributions since the distribution on the intersection between $S^{d-1}$ and the hyperplane does not affect the distribution of $t$. To resolve the concern, we carry out an empirical study of the distribution on the intersection (i.e. conditioned on $t$). Specifically, we train 1000 CNN models with $k=40$ and seeds 1~1000 on CIFAR-10 and compute $t$ regarding each test sample. The distribution of the first sample is visualized in Fig. R7(left). We partition the range of $t$ to 10 intervals by every 10 percent of the frequency, and inspect the **direction of the mean of the gradients in each interval**, each direction is estimated by 100 models. If these conditional mean directions are consistent with the population mean direction, then the gradients are symmetrically distributed on each $S^{d-2}$ hypersphere (R7(right)), thus verifying the rotational symmetry. We investigate the cosine similarities between the conditional and unconditional mean directions on the first 1000 samples. The 10*1000 similarity values have a mean and std at approximately 0.970 and 0.013 respectively. Thus the rotational symmetry is empirically verified.
> - Marginalization Derivations (W3). The marginal distribution $p_{original}$ in Sec. 3.3 is obtained by taking the marginalization of Saw distribution through integration on the hypersphere. It was not claimed as a theorem or lemma since it is not novel. We will include a detailed derivation in the appendix.
>
> **[Gaussian Assumptions (W3, Q2)]**
>
> - The Gaussian distribution described in the appendix is not related to the Saw distribution by any means. The only assumption involved in the Saw distribution is the symmetry over the $S^{d-2}$ (e.g. dashed circles in Fig. 4).
> - Appendix C serves as a sanity check or a null hypothesis on how to interpret the empirically observed cosine similarities. It is introduced in Sec. 3.1 L215, where the cosine similarities between the population means of different model families are studied. Fig. 5 shows that such cosine similarities are very high. However, the significance of these results may not be well interpreted given that in 2-D/3-D scenarios, this cosine similarity range may not be considered very high. Therefore, the analysis in Appendix C aims to uncover how difficult it is to achieve high cosine similarity as the dimension increases (e.g. Fig. 16).
>
> **[Generalizations & Population Means (Q0)]**
>
> It can be observed that as the model capacity increases, the distributions of models become more concentrated (illustrated in Fig. 4(a) and verified in Fig. 3,5,6, etc.). This suggests that **as a single model approaches the population mean, the testing performance also increases**. To verify this claim, we carry out the experiments on deep ensembles since they approach the population mean of the model directly instead of by increasing capacity like single models. It can be found in Fig. 8 that although approaching the population mean differently, deep ensembles and single models have similar trends. This provides support for the connection between the population means and better generalizability.

---

> > ### Comment · Reviewer_We4a · 2024-08-14
> > **Response**
> >
> > Thank you for your response, I hope that the authors can improve the presentation of their theoretical claims, since currently it seems hard to follow.

---

> > > ### Author Response · Authors · 2024-08-14
> > > **Thanks very much for your response**
> > >
> > > We appreciate your comments and suggestions very much! We will revise the manuscript to improve the presentation of the theoretical claim and avoid potential miscommunications.
> > > Regarding the technical concerns, we believe that these have been adequately addressed in our responses. Therefore, with all due respect, we sincerely hope you will reconsider the assessment of our work.
> > >
> > > Kind regards,
> > >
> > > Authors

---

### Official Review · Reviewer_RiRS · 2024-07-15

**Soundness:** 3
**Presentation:** 2
**Contribution:** 3
**Rating:** 7
**Confidence:** 3

**Summary:**

This paper studied the uncertainty introduced in stochastically optimized DNNs via the input saliency maps. By empirical evaluations, the authors discovered that 1) within the same model architecture, models with different capacities tend to align in terms of their population mean directions of the input salience, 2) the distributions of the optimized models concentrated more around their shared population mean as the capacity increases. These observations are very interesting and shed some light on the applications of DNNs such as black-box attacks.

**Strengths:**

The authors took a suitable approach to study the uncertainty introduced in stochastic training of DNNs. The observations made by them are very interesting and inspiring, and well supported by empirical evidences. The authors did a good job explaining their main discoveries using figures and equations. These findings could shed light on the practice of DNNs.

**Weaknesses:**

The main complaint I have on the paper is its presentation. Although the observations quite interesting to me, I found the writing of the paper can be improved.

1. More context is needed for Figure 1.
1.1. How are the dots plotted? I assume you did something like PCA to reduce the gradient dimension to 3 and then plot them on the sphere.
1.2. You said on line 55 that the second hypothesis is illustrated in Figure 1(b), but there is no indication of model capacities in the figure.

2. Even though the authors tried to distinguish between model architecture from model family, the introduce of width and depth complicates the discussion and makes the definition less clear. It would be good to refine the presentation here. For instance, you could give a concrete example like ResNet_depth10_width10_seed0 and ResNet_depth10_width10_seed1 has the same model architecture and belong to the same model family. ResNet_depth10_width10 and ResNet_depth10_width20 has the same model architecture and belong to different model families, etc.

3. It is unclear to me what is defined on line 96.

4. The presentation of the paragraph on line 101 and Figures 2 & 3 can be improved.
4.1. Why do you use the notation $\rho_\text{ind}$? What does ind mean?
4.2. Can you give clear, different names to $\rho_\text{ind}$ and $\rho$, and make both of them display equations?
4.3. In the captions of Figures 2 & 3, can you use the $\rho$ notations and refer to their display equations, respectively?
4.4. The x and y labels in Figures 2 & 3 should be $k_1$ and $k_2$.

5. $\mu(k; x)$ is not defined in Hypothesis I.

6. Some $k_1$ should be $k_2$ on line 158 and in the display equation below. And why do you need $k_1 > k_2$ for this?

**Questions:**

Besides the points I made in the Weakness section, I have one more questions. When practitioners scale their models, they usually increase both depth and width simultaneously (especially in the context of scaling law). In your experiments, you mostly studied the effect of increasing width. Could you add results in the flavor of Figures 2 & 3 where the model capacity is parametrized by both the depth and width following some scaling law relationship?

---

> ### Author Rebuttal · Authors · 2024-08-07
>
> ## Response to Reviewer RiRS
>
> We thank the reviewer very much for acknowledging our work! We now answer the reviewer's questions as follows to make sure that all remaining concerns are resolved.
>
> **[Figure 1 Clarifications (W1)]**
>
> We appreciate the reviewer for pointing out the potential ambiguity of Fig. 1. We clarify that Fig. 1 is for demonstration purposes and to provide a straightforward intuition for the hypotheses and the phenomena. We will update the manuscript to make sure there is no miscommunication. 1.1. The dots are plotted synthetically for a clear visualization. And 1.2. Fig. 1(b) is to illustrate the hypothesis of (i) shared means and (ii) convergence trend compared with Fig. 1(a).
>
>
> **[Model Architecture Clarification (W2)]**
>
> We thank the reviewer very much for the valuable point. For a better presentation, We will add examples to elaborate on how model architectures and families are defined, following the ways suggested by the reviewer.
>
> **[Definition of $\mathcal{F}$ in L96 (W3)]**
>
> L96 aims to define the space $\mathcal{F}$ of functions we focus on in this work.
> Note that given a fixed input dimension, all models can be seen as a function $f:\mathbb{R}^d\rightarrow\mathbb{R}$, regardless of the architecture, the capacity, etc. However, if the space of models $\mathcal{F}$ is simply defined as all such functions
> $\\{f\in\mathcal{C}|f:\mathbb{R}^d\rightarrow\mathbb{R}\\}$, many invalid models will be included. Therefore, we add a constraint to the functions in $\mathcal{F}$ that they fit the training distribution well. We will revise the definition to a mathematical formula like $\mathcal{F} = \\{f\in\mathcal{C}(\mathbb{R}^d)|\mathcal{L}(f;\mathcal{X}\_{train}, \mathcal{Y}\_{train}) < \epsilon \\}$.
>
>
> **[Definitions of $\rho,\rho_{ind}$ (W4)]**
>
> We thank the reviewer for the suggestion! It can indeed be a little confusing between the notations $\rho$ and $\rho_{ind}$. We clarify them as follows.
>
> 4.1. The subscript "ind" in the notation $\rho_{ind}$ refers to "individual". It is defined in L103 as **the cosine similarity of two models**. Note that the inputs of $\rho_{ind}$ are $f^{(1)},f^{(2)}$, which are two functions (models). Differently, $\rho(k_1,k_2)$ (defined in Eq. 1) represents **the expectation of the cosine similarity between two arbitrary models from $\mathcal{F}(k_1)$ and $\mathcal{F}(k_2)$**. It takes the width parameters $k_1,k_2$ as the input. Ideally, $\rho$ should be used to study the similarity. And we carry out the experiments by training 100 models for a given $k\in\\{10,20,40,80,160\\}$ for the four model architectures and three datasets. This already results in $100\times 5\times 4\times 3=6000$ trained models. To demonstrate the influence of the capacity $k$ in a higher resolution, we investigate $k\in\\{8, 10, 12, 14, 16, 20,\cdots, 384, 448\\}$ (L106). This is computationally infeasible for $\rho$. Therefore, we compare the results between $\rho_{ind}$ and $\rho$ in Fig. 6 to verify that **$\rho_{ind}$ serves as a computationally efficient surrogate for $\rho$**.
>
> 4.2. We will term $\rho_{ind}$ and $\rho$ clearly as "individual similarity" and "average similarity" to avoid ambiguity. And as suggested by the reviewer, we will move the definition of $\rho_{ind}$ (L103) and $\rho$ (Eq. 1) together for a better presentation.
>
> 4.3. Originally, Fig. 2 serves as the purpose of motivating the problem and appears before the definition of $\rho, \rho_{ind}$. Thus $\rho_{ind}$ was not used in the caption. As suggested by the reviewer, we will re-arrange the figure and text to include $\rho_{ind}$ in the caption of Fig. 2 for a better presentation.
>
> 4.4. We appreciate the reviewer for pointing this out. We will revise the labels in Fig. 2 and 3.
>
>
> **[$\mathbf{\mu}(k;\mathbf{x})$ in H1 (W5)]**
>
> Thanks for pointing this out! $\mathbf{\mu}(k;\mathbf{x})$ refers to the normalized population mean of the gradient, which is defined in L152. We will re-arrange this part and move the definition to "The Intra-Family Hypothesis." for a better presentation.
>
> **[$k_1,k_2$ in L158]**
>
> Note that in L158, we are discussing the cross-family similarity regarding the population mean of each family. Since it is studied through the cosine similarity, it is symmetric between $k_1$ and $k_2$. Note that for cross-family similarity, $k_1\ne k_2$. Therefore, the condition $k_1>k_2$ is equivalent to the condition $k_1\ne k_2$ in H2. We will revise the manuscript to make it consistent throughout the discussion.
>
>
> **[Depths and Widths (Q1)]**
>
> We appreciate the valuable suggestion from the reviewer. We acknowledge that the depth is also a factor that affects the model capacity. However, since the depth change usually leads to inconsistency in widths, we study the influence of depths by the -small and -large suffixes in the manuscript.
>
> To address the reviewer's question regarding the influence of depths completely, we carry out additional experiments with different settings so that depths and widths can be altered independently by parameters $d$ and $k$, respectively. $d$ determinse the number of layers, and $k$ determines the width of the layers. Unlike the gradually increasing channel (e.g. $[k,2k,4k,8k]$) in the manuscript, here we set the same number of channels for all layers. In this way, the change in the number of layers does not affect the widths anymore. For example, when $d=3$, the layers are $[k, k, k]$. The results are shown in Fig. R1 of the attached PDF file.
>
> It can be found that (1) Given a fixed depth or width, the influence of the other factor is similar when scaled up. (2) Depths are slightly different from widths. Larger widths lead to higher similarities, while closer structures in depths have higher similarities. This is verified by higher similarities near the diagonal entries for Fig. R1 (Right) compared with Fig. R1 (Left).

---

> > ### Comment · Reviewer_RiRS · 2024-08-14
> > **Thank you for the response**
> >
> > I thank the authors for their detailed response. I have no more questions.

---

### Official Review · Reviewer_fWRt · 2024-07-16

**Soundness:** 3
**Presentation:** 2
**Contribution:** 2
**Rating:** 6
**Confidence:** 2

**Summary:**

This paper studies the distribution of input saliency maps of trained neural networks at varying depth and width. The authors observe that as the model capacity increases, these distributions converge towards and become concentrated around a shared mean direction. The authors also state two hypotheses for the behavior of these distributions and provide empirical evidence for them.

**Strengths:**

- The paper clearly formulates and discusses two hypotheses (H1,H2) that help capture the empirical observations.
- While the experiments are small scale, they are sufficient to support the authors main claims and observations.

**Weaknesses:**

- The authors do not distinguish between the stochasticity introduced by random initialization and the stochasticity introduced by optimization. In Appendix A, they discuss the similarity scores at initialization, but a more interesting ablation would be to compare fixing the initialization and then randomizing the batches with fixing the ordering of the batches but randomizing the initialization.
- Related to the above point, there is no discussion of the effect of the learning rate or batch size which should greatly affect the final results. This would help disentangle the randomness introduced by initialization vs optimization.

**Questions:**

- Have the authors explored the effects of different optimization hyperparameters (e.g. learning rate, batch size) or optimization algorithms (e.g. Adam) on the convergence trends observed in the paper?
- The experiments in this paper are focused on computer vision. Do the authors expect similar conclusions to hold in other domains as well (e.g. NLP)?

**Limitations:**

The authors have adequately addressed the limitations of their work.

---

> ### Author Rebuttal · Authors · 2024-08-07
>
> ## Response to Reviewer fWRt
>
> We appreciate the reviewer very much for the acknowledgment of our work. To further strengthen the reviewer's confidence in our findings, we address the reviewer's remaining questions as follows.
>
> **[Fixed Initializations with Random (W1)]**
>
> We thank the reviewer for the suggestion of separating the stochasticity of random initialization of the model weights and the stochasticity of the randomization of batch orders in SGD.
>
> In order to address the reviewer's concern, we carry out additional experiments, where models are all initialized with the same weights by setting `seed = 0` before the initialization. After the initialization, we set again manually to 1~100 in the training process. The experiments are carried out using CNNSmall models with $k=20$ on CIFAR-10.
> *It should be noted that this is actually equivalent to uniformly sampling from the original distributions. Therefore, we expect an almost identical distribution compared with the models used in the manuscript, where seeds are determined at the beginning of all steps.* Since the normalized gradients $\mathbf{u}$ are of 3072 dimensions, it is infeasible to compare the distribution directly. To verify the uniformity, we inspect $t = \mathbf{u}^T\mathbf{\mu}(\mathbf{x})$ described in L177-190 of the manuscript. This marginal distribution is 1-D on $[-1,1]$ and we study the Wasserstein distance between the distribution of models with identical (random) initialization and the distribution of models with different (random) initializations. The distance is approximately $0.0225$, which means that the distributions are very close in terms of the dispersion degree.
>
>
> **[Other Sources of Stochasticity (W2, Q1)]**
>
> We agree with the reviewer that different training criteria such as the learning rates, the batch sizes, the solvers, etc. can affect the resulting models. However, it should be noted that a criterion determines a distribution of trained models. And the phenomenon discovered in this work holds for all kinds of criteria instead of the one used in the manuscript.
>
> To completely resolve the reviewer's concern, we carry out additional experiments to investigate the influence of different training options, including learning rates, batch sizes, solvers, and data augmentations. The experiments are described in the global response in detail, and the results are presented in the attached pdf file.
>
> **[Other Data Types]**
>
> We thank the reviewer very much for the question regarding other data types. We clarify that text data is not commonly studied in the literature on benign overfitting. Additionally, conventional NLP training is typically conducted in an unsupervised manner, making it difficult to carry out similar experiments. However, since NLP data are now often trained using transformers, we investigate whether the transformer and attention mechanism exhibit this property as a starting point.
>
> Specifically, we train vision transformer (ViT) models on CIFAR-10 with varying capacities controlled by $k\in \\{10, 20, 40, 80\\}$. CIFAR-10 has an input size of $32\times 32$ pixels, thus we set patch size to be $4\times 4$, resulting in $8\times 8$ patches. We set the embedding size to $4k$, divided by $k/2$ heads, and we set the depth to $8$. We then vary seeds in 1~100 and train 100 models of each $k$ and study the mean of the similarity $\rho(k_1,k_2)$ (i.e. the same experiments as Fig. 3 in the manuscript) and the similarity of the population mean (i.e. the same as Fig. 5 in the manuscript). The results are shown in Fig. R6.
> It can be observed that although distinct from convolutional layers, the transformer structure also has the discovered convergence trend. It can also be noted that the degree of dispersion of ViTs is much higher than CNNs.
>
> Therefore, based on the additional experiments on transformers, we can deduce that the convergence trend may also hold for NLP tasks where transformers are extensively used.

---

> > ### Comment · Reviewer_fWRt · 2024-08-12
> >
> > I thank the authors for addressing my questions and concerns. I still have a few remaining questions but in response to the rebuttal, I have raised my score.
> >
> > > It should be noted that this is actually equivalent to uniformly sampling from the original distributions. Therefore, we expect an almost identical distribution compared with the models used in the manuscript, where seeds are determined at the beginning of all steps.
> >
> > I agree that each individual run has the same distribution, but the runs are now correlated. As an extreme, if I match both the initialization and batch-sampling random seeds, the runs will again also be identically distributed but will be fully correlated. My question was specifically whether the differences between the saliency maps of models at finite size were caused by the stochasticity from initialization, or from the randomness in the batches. For example, it seems a priori possible that for each initialization, the distribution has a *different mean* (depending on the initialization) and a *smaller variance,* but then when you randomize the initialization, these distributions are mixed together creating an overall distribution with a new mean and a *larger variance.*
> >
> > > [Other Sources of Stochasticity (W2, Q1)]
> >
> > I thank the reviewer for the additional experiments, and they are a valuable contribution to the overall story. It appears that the batch size makes little to no difference, but the learning rate and optimization algorithm make a fairly large difference (which is to be expected). Indeed, the first observation suggests that the randomness is almost entirely due to the random initialization. I think it would be extremely interesting to include a cosine similarity plot (width x width) where the first distribution uses a fixed initialization (seed=0) and a random ordering of the batches, and the second distribution uses another fixed initialization (seed=1) and a random ordering of the batches. This could also be flipped so that you fix two orderings of the batches and then randomize over the initializations for each distribution. This would help disentangle these two effects.

---

> > > ### Author Response · Authors · 2024-08-14
> > > **Thank you for your positive feedback**
> > >
> > > We appreciate the reviewer for the timely response to the rebuttal. Regarding the reviewer's remaining questions about the random initializations, we carry out additional experiments accordingly.
> > > We start by formalizing this problem for clarity:
> > >
> > > **Problem Setup:** Given a training scheme and model family $\mathcal{F}(k)$, the training procedure leads to a distribution of trained models $p(f)$. As we agreed, when the initialization is fixed to $\theta$, it is the training procedure is essentially sampling from the conditional distribution $p(f|\theta)$ instead of the unconditional distribution $p(f)$.
> > >
> > > **Original Approach:** In the original rebuttal, we studied the Wasserstein distance between the unconditional distribution $p(f)$ and the conditional distribution $p(f|\theta_0)$. The latter one is achieved by fixing the initialization with `seed=0` and changing the seed from 1 to 100 in the training procedure. We found out that the distributions of $t$ is distributed very similarly with or without the conditions.
> > >
> > > **Comparison between Conditional Distributions:** As suggested by the reviewer, we focus on two conditional distributions $p(f|\theta_0)$ and $p(f|\theta_1)$, where $\theta_1$ represents the initializations under `seed=1`. Other settings are identical to $p(f|\theta_0)$. We thus have $f^0_1,\cdots,f^0_{100}\sim p(f|\theta_0)$ and $f^1_1,\cdots,f^1_{100}\sim p(f|\theta_1)$. The superscript indicates the initialization seeds and the subscript indicates the training seeds.
> > >
> > > - **Individual Similarity**
> > >
> > > First, we notice immediately that the training seeds for both $\theta_0$ and $\theta_1$ are 1~100. This means that $\forall i, f^0_i, f^1_i$ differ only in initializations. We inspect (a) $\rho_{ind}(f^0_i, f^1_i)$ (100 pairs) to see if they have exceptional similarity compared with (b) $\rho_{ind}(f^0_i, f^1_j), i\ne j$ ($\binom{100}{2}=4950$ pairs). Besides, within the same condition, all models only differ in terms of the orders of the training batch. We thus also inspect the similarity of all models of the same condition: (c) $\rho_{ind}(f^0_i,f^0_j), i\ne j$ and (d) $\rho_{ind}(f^1_i,f^1_j), i\ne j$. Each of them has $\binom{100}{2}=4950$ pairs.
> > >
> > > As demonstrated in Tab. R1, (i) the comparison between (a)(b) indicates that with different inializations, the same order of batches in the training procedure does not contribute to higher similarities. (ii) The comparison among (b)(c)(d) indicates that the same initialization indeed leads to higher similarity even though the order of batches is distinct. This corresponds to the reviewer's intuition. It should be noted that the contributions of batch orders and initializations are also affected by the number of epochs. Intuitively, more training epochs should lead to smaller contributions from the initializations but greater contributions from the batch orders. We will add experiments in the manuscript to explore these factors.
> > >
> > > Items|(a) Diff. Init. Same Order|(b) Diff. Init, Diff. Order|(c) Same Init. $\theta_0$ Diff Order|(d) Same Init. $\theta_1$ Diff Order
> > > -|-|-|-|-|
> > > \# of pairs|100|4095|4095|4095|
> > > mean of $\rho_{ind}$|0.0758|0.0753|0.0879|0.0855|
> > > std. of $\rho_{ind}$|0.0038|0.0037|0.0042|0.0048|
> > >
> > > **Table R1:** The comparison between the similarities between single models of different criteria.
> > >
> > > [Continued in the next block]

---

> > > > ### Author Response · Authors · 2024-08-14
> > > > **[Continued Response]**
> > > >
> > > > - **Mean Similarity**
> > > >
> > > > Now we study the population mean of $p(f|\theta_0)$ and $p(f|\theta_1)$. We admit that reimplementing the results of Figure 5 for these distributions requires training $200\times|\\{10,20,40,80,160\\}|=1000$ models, which is infeasible given the limited time. We control the number of models used to approximate the population mean by $M_0,M_1\in\\{1,2,5,10,20,50,100\\}$. As the number increases, the approximation becomes closer to the population mean. And we investigate $$CosSim(\nabla\tilde{f}^0_{M_0}(x)),\nabla\tilde{f}^1_{M_1}(x))$$, where $\tilde{f}^l_M(x) = \frac{1}{M}\sum_{i=1}^Mnorm(\nabla f^l_k(x)),l=0,1$ is the average of the normed empirical mean estimated by $M$ models. This leads to a $7\times7$ heatmap, shown below as a table in Tab. R2.
> > > > As demonstrated in Fig. 8, increasing $M$ and increasing $k$ change the approximation to the population mean in a very similar way. The following results demonstrate that for models with **distinct initializations but the same batch order**, the conclusion still holds, and the values are almost the same as the similarity between unconditional CS-20 models shown in the top left corner of Fig. 5(a).
> > > >
> > > > |M0\M1|1|2|5|10|20|50|100|
> > > > |-|-|-|-|-|-|-|-|
> > > > |1|0.07|0.10|0.15|0.18|0.21|0.23|0.24|
> > > > |2|0.10|0.14|0.20|0.24|0.28|0.31|0.33|
> > > > |5|0.14|0.19|0.28|0.34|0.39|0.44|0.46|
> > > > |10|0.17|0.24|0.34|0.41|0.48|0.53|0.56|
> > > > |20|0.19|0.27|0.39|0.48|0.55|0.62|0.65|
> > > > |50|0.22|0.30|0.44|0.54|0.62|0.70|0.74|
> > > > |100|0.22|0.32|0.46|0.56|0.65|0.73|0.77|
> > > >
> > > > **Table R2:** The similarity between the empirical mean of $p(f|\theta_0),p(f|\theta_1)$. $M_0,M_1$ represents the number of models used to estimate the empirical mean.
> > > > For the models with the **same initialization but different batch orders**, we can carry out the same experiments. For $p(f|\theta_0)$, we split the seeds into $1\sim50$ and $51\sim100$ to estimate the population mean, respectively. Note that for $p(f|\theta_1)$, the results are almost identical to $p(f|\theta_0)$ and thus omitted here. The results are shown in Tab. R3, where similarities are higher than Tab. R2. This verifies that initializations have higher contributions to the stochasticity compared with the batch orders.
> > > >
> > > > |M0\M0|1|2|5|10|20|50|
> > > > |-|-|-|-|-|-|-|
> > > > |1|0.08|0.12|0.17|0.20|0.23|0.26|
> > > > |2|0.12|0.16|0.23|0.28|0.32|0.36|
> > > > |5|0.16|0.23|0.32|0.39|0.45|0.51|
> > > > |10|0.20|0.28|0.39|0.48|0.55|0.62|
> > > > |20|0.23|0.32|0.45|0.55|0.64|0.72|
> > > > |50|0.26|0.36|0.51|0.62|0.72|0.81|
> > > >
> > > > **Table R3:** The similarity between the empirical mean of $p(f|\theta_0)$, estimated by different groups of models.
> > > >
> > > > We thank the reviewer very much for pointing out this insightful factor. We will update these results in the manuscript with more comprehensive experiments and a more detailed discussion.

---

### Official Review · Reviewer_YpLK · 2024-07-17

**Soundness:** 2
**Presentation:** 3
**Contribution:** 2
**Rating:** 5
**Confidence:** 4

**Summary:**

This paper investigates the distribution of trained deep neural networks through the lens of input saliency maps. It empirically shows that as the capacity of either of two stochastically optimized models increases, they tend to resemble each other more. Therefore, the authors hypothesize that within the same architecture (Resnet or CNN), different variants align in their input salience directions, and converge towards a shared population mean as their capacity increases. Furthermore, they propose a semi-parametric model, based on the Saw distribution, to capture this convergence trend. These findings enhance the understanding of various applications such as black-box attacks and deep ensembles.

**Strengths:**

1. The paper is well-written.
2. The experiments are comprehensive and well-designed.
3. The input saliency maps perspective on the ensemble problem is novel.

**Weaknesses:**

1. The primary finding is not entirely novel. Previous research has demonstrated that the performance of neural network ensembles converges to that of a single model as the model size increases [1,2]. Similarly, as the size of the models increases, the variance in predictions and disagreement among ensemble members diminishes [1].
2. Several details about the experimental setup are omitted. These include the optimizer used, learning rate, learning rate schedule, batch size, number of iterations, weight decay, and data augmentation techniques.
3. The choice of optimizers and hyperparameters can significantly alter the distribution of trained models. However, the study lacks an analysis of their findings across different optimizers, hyperparameters and sources of stochasticity.
4. The experiments are conducted in small-scale settings, which may limit the generalizability of the results.

[1] Geiger, Mario, et al. "Scaling description of generalization with number of parameters in deep learning." _Journal of Statistical Mechanics: Theory and Experiment_ 2020.2 (2020): 023401.


[2] Kobayashi, Seijin, Johannes von Oswald, and Benjamin F. Grewe. "On the reversed bias-variance tradeoff in deep ensembles." ICML, 2021.

**Questions:**

1. Have the authors conducted experiments to validate their findings across different optimizers and related hyperparameters?
2. Stochasticity can be introduced in the training of Neural Networks in various ways. Changing the initialization through different seeds is one possibility but other possibilities include for example training on different subsets of the data, large or small batch sizes, large or small learning rates, data augmentation or label noise. Have the authors validated their findings for different sources of stochasticity?
3. In Figure 8, why does the Large ResNet exhibit a higher test loss than the small CNN?
4. The explanation for the effectiveness of deep ensembles suggests that they better approximate the population mean. However, this assumes that the mean corresponds to a good generalizing model. Could the authors provide evidence or an explanation to support this assumption?
5. The introduction mentions "input salience" several times without providing a definition. For better clarity and understanding, it would be helpful to include a definition of "input salience" in the introduction.
6. In section 2.1, the authors state, "We focus on $f: \mathbb{R}^d \to \mathbb{R}$ which predicts the logit specifically for the targeted class." Could the authors clarify this statement? Does it mean that the analysis after training considers only the logit corresponding to the true class?
7. In section 2.2, $\mathbf{u}_1$ and $\mathbf{u}_2$ are sometimes used to represent different samples from the same set $\mathcal{G}_k$, and sometimes from different sets.  It would be better to use a different notation to differentiate between the two cases.
8. In Hypothesis I, line 147, the term $\mu(k;x)$ is only introduced later in the text.
9. In line 234, could the authors clarify the definition of $\rho_{ind}(f,f;x)$ and explain its relationship with $\rho(k,k;x)$ ?

**Limitations:**

The limitations of the study are correctly addressed in the manuscript.

---

> ### Author Rebuttal · Authors · 2024-08-07
>
> ## Response to Reviewer YpLK
> We thank the reviewer for the invaluable feedback. We address the questions as follows.
>
> **[Novelty (W1)]**
>
> We summarize the work in [5,6] and how the novel contributions of our work on deep ensembles differ from them. It should also be noted that exploring the mechanism of deep ensembles is not our main contribution, but an application and verification of our work.
> - [5] studies the failure modes of deep ensembles and. However, [5] focuses on empirical observations of the performance of ensembles. This is never claimed as our contribution. Note that the comparison between the **performance** of single models and ensembles is not novel and can be found in many works discussed in our manuscript (e.g. [7]). However, the reason behind deep ensembles' performance gain is still mysterious [5]. Our work hypothesizes and verifies that (1) single models' performance gain is related to the convergence trend, and (2) the convergence is towards the shared mean of **all capacities**. This provides a novel perspective to understand deep ensembles and why an ensemble of many *small* models can achieve the performance of a single *large* model.
> - [6] studies the double descent phase of DNNs using NTKs. The analysis based on NTK verifies similar results for deep ensembles as in [5]. As a result, the novelties of our work discussed above still hold. The similarity of mechanisms between models with different capacities and how the convergence is towards the shared population means across capacities are also novel compared with existing work. Additionally, NTK is a powerful tool yet with limitations and strong assumptions. The analysis is on the global loss regarding the data distribution (Eq. 4 in [6]). However, our results are on the direct prediction of any single input and its neighbors.
>
> We will add these works to the related work section for a more comprehensive review of existing work.
>
> **[Experiment Setups (W2)]**
>
> To stay consistent across experiments, we follow the setups of [1] as claimed in L195-197. We use SGD solver with a learning rate $\gamma = 0.1/\sqrt{1+epoch}$. No weight decay, momentum, or data augmentation are included to minimize variants. The batch size is 128. All models are trained for 200 epochs.
>
> **[Sources of Stochasticity (W3, Q1, Q2)]**
>
> Training schemes do not affect the discovery. Please refer to the global response for details, including comprehensive additional experiments.
>
> **[Scale of Experiments (W4)]**
>
> This work is mostly based on the benign overfitting and double descent phenomena (e.g. [1-4]), where large models counterintuitively do not exacerbate overfitting. We aim to provide insights into this mysterious capability of DNNs from the perspective of XAI. Hence the settings of our work are mostly based on existing studies on this topic, too.
>
> The complexity of the datasets used in our experiments is a strength, not a weakness. Unlike existing works, which mainly use CIFAR (e.g. [1]), we test our hypotheses on TinyImageNet, which includes 200 classes and 110k samples, offering a more complex and realistic data distribution. The same trends observed in CIFAR datasets are also evident in TinyImageNet, suggesting these trends will hold for even more complex datasets like ILSVRC.
>
> We admit that carrying the experiments over ILSVRC can be infeasible for the studies of benign overfitting. As explained in Appendix B, with $k=64$, ResNetLarge is already equivalent to ResNet-18. We scale $k$ up to 448, which makes it impossible to train a group of models to estimate the empirical mean. For example, in [4], a study of overfitting carried out experiments on CIFAR and SVHN.
>
> **[Deep Ensemles and Single Models (Q4)]**
>
> As the capacity increases, the distribution becomes more concentrated (illustrated in Fig. 4(a), verified in Fig. 3,5,6, etc.). This suggests that *as a model approaches the population mean, the performance increases*. To verify this, we carry out the experiments on deep ensembles since they approach the population mean of the model directly instead of by increasing capacity like single models. Fig. 8 shows that although approaching the mean differently, deep ensembles and single models have similar trends. This provides support for the connection between the population means and better generalizability.
>
> **[Clarifications]**
> - Q3 [ResNet]. ResNets have higher expressiveness compared with CNNs and rely heavily on data augmentations. Since model performance is beyond our scope, data augmentation is not included. Fig. R5 shows additional results of ResNets with data augmentations. It improves the performance of ResNets greatly. However, the convergence trend is not affected.
> NLL is also affected by confidence. ResNets tend to be more confident in wrong predictions, leading to higher losses.
> - Q5 [Input Salience]. Input salience is usually used by the XAI community to refer to the input gradient as a saliency map, and also used to refer to attribution-based methods. We will formally introduce the definition of the concept.
> - Q6 [Logit]. Logits refer to the prediction of the target class before softmax. We also include the results of the post-softmax output in Appendix A. It shows that this does not affect the results. Note that taking the gradient of the output probability of the target class is equivalent to the gradient regarding the negative log-likelihood, normalized by the probability.
> - Q7 [$u_1,u_2$]. We will use $u_1,u_2$ to refer to gradients of different sets and $u,v$ to refer to those of the same sets.
> - Q8 [$\mu(k;x)$]. We will move the definition of $\mu(k;x)$ to H1 for a better presentation.
> - Q9 [$\rho_{ind}$]. $\rho$ is defined in eq. (1). It refers to the similarity between models of capacity $k$ and is approximated by taking the average over 100 models each. $\rho_{ind}$ refers to the model-dependent similarity computed regarding the two models (in L233). We will define it more clearly to avoid ambiguity.

---

> > ### Comment · Reviewer_YpLK · 2024-08-13
> >
> > Thank you for the provided clarifications and additional experiments. However, I still have the following concerns:
> >
> > 1. Based on your explanation, if I understand correctly, the hypothesis is that the population mean is consistent across different capacities for a given architecture and that both ensembling and increasing capacity are methods to approach this population mean (that is why ensembling works). Could you clarify how your experiments support the conclusion that both methods (ensembling and increasing capacity) converge to the same population mean? Is it only based on the fact that they share similar scaling for the test Loss? Have the authors tried to measure similarities between the ensemble output and single but larger models?
> >
> > 1. I still have some concerns regarding the experimental setup. For instance, in the newly provided experiments, specifically in Figure R5 (top right, there seems to be a mismatch between the plot title and the y-axis label) it appears that the maximum test accuracy on CIFAR-10 with ResNet, across all widths, is capped at 0.5, which is highly unusual. Similarly, in the experiments comparing Adam and SGD, it seems that Adam achieves better test error compared to SGD, which is generally unexpected see for example [1]. Could you provide further clarifications?
> >
> > [1] Wilson, Ashia C., et al. "The marginal value of adaptive gradient methods in machine learning." Advances in neural information processing systems 30 (2017).

---

> > > ### Author Response · Authors · 2024-08-14
> > > **Thank you for your valuable comments**
> > >
> > > We appreciate the reviewer very much for your comments. This allows us to resolve the reviewer's remaining concerns further.
> > >
> > > **[Single Models & Ensembles]**
> > >
> > > > The hypothesis is that the population mean is consistent across different capacities for a given architecture and that both ensembling and increasing capacity are methods to approach this population mean
> > >
> > > Yes, this understanding regarding the hypothesis about ensembles is correct.
> > >
> > > > Could you clarify how your experiments support the conclusion that both methods (ensembling and increasing capacity) converge to the same population mean?
> > >
> > > - **Ensemples**:
> > >
> > > Note that for homogenuous ensembles, all enesemble members $f^{(1)},\cdots,f^{(M)}\in\mathcal{F}(k)$ are from the same family (i.e. same capacity $k$). Therefore, their mean is by definition the approximation to the population mean of the entire family. Note that the error of such an approximation is inversely controlled by the number $M$ of ensemble members, we train $M=100$ ensemble members to minimize the error as much as possible. Therefore, the following statement
> > >
> > > > the population mean is consistent across different capacities
> > >
> > > is verified by comparing the approximated population mean of different families $\mathcal{F}(k_1), \mathcal{F}(k_2)$. The results are shown in Fig. 5 of the manuscript. It can be observed that the cosine similarity even reaches 0.9, which is significant given the curse of dimensionality.
> > >
> > > - **Single Models**:
> > >
> > > Unlike ensembles, single models $f\in\mathcal{F}(k)$ approach the population mean as $k$ increases. This is demonstrated in our experiments. For example, Fig. 3 shows the expected similarity of two randomly picked models of capacity $k_1,k_2$. Larger $k$s lead to higher expected similarity between single models. This suggests a decreasing dispersion as $k$ increases. Therefore, the following statement is verified.
> > >
> > > > both ensembling and increasing capacity are methods to approach this population mean
> > >
> > > For ensembles, we have explained above that it approaches the population mean by definition. As for single models, they approach the population mean as $k$ increases due to the decreasing dispersion.
> > >
> > > - **Performance**:
> > >
> > > Now that we have established that *both ensembles and single models can approach the population mean by increasing $M$ and $k$ respectively*, we carry out experiments in Fig. 8 to demonstrate **whether the distance to the population mean is related to the performance**. Here we also purposely align the x,y-axes of the two subfigures. It is observed that even though ensembles and single models approach the population mean differently, the testing performance (y-axis) is related to the dispersion degree (x-axis) in the same pattern.
> > >
> > >
> > > **[Clarification on Fig. R5]**
> > >
> > > We sincerely appreciate the reviewer for pointing this out so that we can clarify the typos of this figure. First, in the right subfigure of Fig. R5, the title should be "Loss" for the left column and "Accuracy" for the right column. As for the accuracy, we clarify that the top row should be the results of **CIFAR-100**, and the bottom row should be the results of **CIFAR-10**. Therefore, the ~50% accuracy is achieved by ResNet trained on CIFAR-100 without data augmentations. As for CIFAR-10 (bottom row), a ResNet trained from scratch with data augmentation achieves an accuracy of ~90%, which is a standard result. We apologize for the typo and the confusion.
> > >
> > > [Continued in the next block]

---

> > > > ### Author Response · Authors · 2024-08-14
> > > > **[Continued Response]**
> > > >
> > > > **[Different Solvers]**
> > > >
> > > > We clarify that the test performance can be affected by many factors. Such differences are both beyond the scope of our work and do not affect any conclusions in our work. We elaborate on this as follows
> > > >
> > > > - **The Many Factors:**
> > > >
> > > > The testing performance can be affected by batch sizes (e.g. Fig. R2), data augmentations (e.g. Fig. R5), learning rates (e.g. Fig. R3), solvers (e.g. Fig. R4), and also other parameters of the solvers (e.g. weight decay, learning rate decay, momentum, etc.). The choices of these parameters create exponentially many possible combinations, and these combinations can also have different effects for different model architectures. **With different combinations, Adam and SGD can outperform each other respectively.**
> > > >
> > > > - **Results in [1]:**
> > > >
> > > > **[1] aims at studying the effectiveness of different solvers in testing generalizations**. Therefore, [1] pushes the limit of each solver by employing complicated and customized schemes to determine the learning rates and decays for different solvers (Sec. 4.1 & App. D). For Adam, it is also claimed in the reference [1] that
> > > >
> > > > >  we find that tuning the initial learning rate and decay scheme for Adam yields significant improvements over its default settings in all cases.
> > > >
> > > > As for SGD, [1] employs a multistep scheduler where both the initial learning rate $0.5$ and the learning rate decay schemes are studied extensively. It is observed in Fig. 1 of [1] that the loss of SGD decreases at distinct rates in different periods due to the scheduler scheme. If we extend one of the steps along its original slope, a SGD model that underperforms Adam model will be obtained, which verifies our previous claim that the performance comparison greatly depends on the training schemes.
> > > > This approach can also create a risk of instability for models of extremely varying capacities. Our work applies a more conservative scheme that scales the learning rate smoothly so that it can be applied to all model architectures and capacities.
> > > >
> > > > Note that from the slow convergence speed in Fig. 1 of [1], we also deduce that data augmentation is applied in [1], which is not included in our experiments. It should be noted that without data augmentation, a learning rate starting at 0.5 can be very risky regarding the overfitting issue. We train CNNSmall models with $k=64$ under SGD w/ the scheme suggested in [1] w/o data augmentation. After 200 epochs, the testing accuracy is only 77.1%. This also verifies the importance of the combinations of parameters regarding the performance.
> > > >
> > > > - **The Unnecessity to Push the Performance Limit:**
> > > >
> > > > Different from [1], in our work, **we study Adam and SGD as two different sources of stochasticity**. Therefore, the very complicated training schemes are beyond the scope of our work. Instead, our experiments have verified that neither the empirical results nor the results are affected by such differences.
> > > >
> > > >
> > > > [1] Wilson, Ashia C., et al. "The marginal value of adaptive gradient methods in machine learning." Advances in neural information processing systems 30 (2017).

---

> > > > > ### Comment · Reviewer_YpLK · 2024-08-14
> > > > >
> > > > > Thank you for the last clarifications; they helped me gain a better understanding of your work and results. I will update my score accordingly.

---

### Author Rebuttal · Authors · 2024-08-07

## Global Response to all Reviewers

We appreciate the reviewers very much for the invaluable feedback and insightful suggestions! First, we address the questions shared by reviewers as follows. All images are shown in the attached PDF file, indexed by the prefix `R` (e.g. Fig. R1).

##Different Sources of Stochasticity

This work aims to reveal the convergence trend of the distribution of model behaviors under the stochasticity of the training criterion. This does not limit the conclusion to the specific criterion used in the manuscript. Distinct training criteria can lead to different distributions of trained models. But these different distributions of trained models **all** satisfy the revealed trend.

**Training Details in the Manuscript**

For the sake of consistency, in this work, we follow the training criterion suggested in [1]. Specifically, stochastic gradient descent (SGD) is used as the solver, with a batch size of 128. The input data are normalized, but not augmented. We start with the initial learning rate $\gamma_0=0.1$ and update it with $\gamma_t=\gamma_0/\sqrt{1+t}$, where $t$ is the epoch.

**Additional Results**

To resolve the reviewer's concern, we present additional experiments to investigate (1) depths and widths; (2) learning rates; (3) batch sizes; (4) solvers. The results are shown in the attached pdf file. The main results should be the similarity *within* each criterion. But we also present the cross-criterion similarity results (e.g. between models trained using different learning rates/solvers/batch sizes/etc.).

We elaborate on them more as follows

- **Depths and widths**: The scale of depths is not as straightforward as width since modifying depths may change widths as well. Therefore, in the manuscript we study the influence of depth by setting -small and -large variations (L809). Here we present additional results that study the influence of depths continuously, with 1~5 layers, each of which is followed by a maxpool layer with stride 2. Finally, an adaptive pooling layer is appended at the end. To rule out the influence of widths (channels), all layers have the same width, determined by $k$. e.g., For the 4-layer scenario, the intermediate layers have widths [k, k, k, k] instead of [k,2k,4k,8k] in the manuscript. The results are shown in Fig R1. It can be found that (1) Given a fixed depth or width, the influence of the other factor is similar when scaled up. (2) Depths are slightly different from widths. Larger widths lead to higher similarities, while closer structures in depths have higher similarities. For widths (left), the similarity always increases left-to-right and top-to-bottom. But for depths (right), pairs near the diagonal have higher similarities.

- **Batch Sizes**: We investigate the influence of batch sizes, varying in {64, 128, 256, 512}. The results are shown in Fig. R2. It can be observed that although different batch sizes lead to different performance (e.g. testing accuracy), the convergence trend holds in all scenarios.

- **Learning Rates**: We test different learning rates on how they affect the results. We include {1e-1, 1e-2, default}, where "default" refers to the criterion used in the manuscript. As shown in Fig. R3, the revealed trend is preserved in all learning rates. It is also worth noticing that learning rates affect ResNets more than CNNs.

- **Solvers**: Apart from SGD, we include Adam, AdamW, and SGD w/ momentum. For Adam and AdamW we set the learning rate to 1e-3, while SGD w/ momentum uses a learning rate of 1e-1 with a momentum of 0.9. The results are shown in Fig. R4. Although different solvers lead to models of different performances, they all preserve the same convergence trend.

- Note that as studied in Fig. 6 of the manuscript, $\rho_{ind}$ can be a computationally efficient compromise to $\rho$. Therefore, we studied $\rho_{ind}$ in these additional experiments. Therefore, for the similarity of the same criterion, we only plot the upper triangular part of the similarity maps.

In conclusion, although training schemes can affect the resulting distributions of models, the influence of the model capacity stays invariant across different criteria. We will include the discussions regarding these variants of training criteria in the manuscript.



### **References**

**(The references of individual responses are also listed here)**

[1] Nakkiran, P., Kaplun, G., Bansal, Y., Yang, T., Barak, B., and Sutskever, I. (2021). Deep double descent: Where bigger models and more data hurt. Journal of Statistical Mechanics: Theory and Experiment, 2021(12):124003.

[2] Cao, Y., Chen, Z., Belkin, M., & Gu, Q. (2022). Benign overfitting in two-layer convolutional neural networks. Advances in neural information processing systems, 35, 25237-25250.

[3] Li, Z., Zhou, Z. H., & Gretton, A. (2021). Towards an understanding of benign overfitting in neural networks. arXiv preprint arXiv:2106.03212.

[4] Mallinar, N., Simon, J. B., Abedsoltan, A., Pandit, P., Belkin, M., & Nakkiran, P. (2022). Benign, Tempered, or Catastrophic: A Taxonomy of Overfitting. arXiv e-prints, arXiv-2207.

[5] Geiger, M., Jacot, A., Spigler, S., Gabriel, F., Sagun, L., d’Ascoli, S., ... & Wyart, M. (2020). Scaling description of generalization with number of parameters in deep learning. Journal of Statistical Mechanics: Theory and Experiment, 2020(2), 023401.

[6] Kobayashi, S., von Oswald, J., & Grewe, B. F. (2021, July). On the reversed bias-variance tradeoff in deep ensembles. ICML.

---

### Author Response · Authors · 2024-08-12
**We are happy to answer more questions if there still exist concerns for our paper.**

Dear Reviewers,

Thanks very much for the time and effort that you have dedicated to reviewing our paper. We greatly appreciate your constructive comments and hope that our response adequately addresses your concerns.

Should you have any further questions or confusion, we are more than willing to provide additional clarifications. Thank you again for your valuable insights.

Best regards,

Authors

---

### Decision · Program_Chairs · 2024-09-25

**Decision:**

Accept (poster)

**Comment:**

This paper examines the the distribution of trained deep neural networks from the perspective of input saliency maps. In particular, paper provides insightful observations regarding how models tend to resemble each other more as their capacity increases and its implications. The reviews for the paper are mostly positive. I recommend acceptance and encourage the authors to address the reviewer's comments in the final version of the paper.